# Non-geodesically-convex optimization in the Wasserstein space

**Hoang Phuc Hau Luu**  **Hanlin Yu**  **Bernardo Williams**  **Petrus Mikkola**
**Marcelo Hartmann**  **Kai Puolamäki**  **Arto Klami**

Department of Computer Science, University of Helsinki

## Abstract

We study a class of optimization problems in the Wasserstein space (the space of probability measures) where the objective function is *nonconvex* along generalized geodesics. Specifically, the objective exhibits some difference-of-convex structure along these geodesics. The setting also encompasses sampling problems where the logarithm of the target distribution is difference-of-convex. We derive multiple convergence insights for a novel *semi Forward-Backward Euler scheme* under several nonconvex (and possibly nonsmooth) regimes. Notably, the semi Forward-Backward Euler is just a slight modification of the Forward-Backward Euler whose convergence is—to our knowledge—still unknown in our very general non-geodesically-convex setting.

## 1 Introduction

Sampling and optimization are intertwined. For example, the (overdamped) Langevin dynamics, typically considered a sampling algorithm, can be considered as gradient descent optimization where a suitable amount of Gaussian noise is injected at each step. There are also deeper connections. At the limit of infinitesimal stepsize, the law of the Langevin dynamics is governed by the Fokker-Planck equation describing a diffusion over time of probability measures. In the seminal paper [39], Jordan, Kinderlehrer, and Otto reinterpreted the Fokker-Planck equation as the gradient flow of the functional relative entropy, a.k.a. Kullback-Leibler (KL) divergence, in the (Wasserstein) space of finite second-moment probability measures equipped with the Wasserstein metric. The discovery connects the two fields and encourages optimization in the Wasserstein space, even conceptually, as it directly gives insight into the sampling context. Studies in continuous-time dynamics [21, 12, 66, 30] seem natural and enjoy nice theoretical properties without discretization errors. Another line of research studies discretization of Wasserstein gradient flow by either quantifying the discretization error between the continuous-time flow and the discrete-time flow [39, 67, 27, 23, 28] or viewing discrete-time flows as *iterative optimization schemes* in the Wasserstein space [65, 26, 70, 11] where the primary focus is on (geodesically) *convex* optimization problems.

Nonconvex, nonsmooth optimization is challenging, even in Euclidean space, quoting Rockafellar [63]: *"In fact the great watershed in optimization isn't between linearity and nonlinearity, but convexity and nonconvexity."* The landscape of nonconvex problems is mostly underexplored in the Wasserstein space. In the sampling language, it amounts to sampling from a *non-log-concave* and possibly *non-log-Lipschitz-smooth* target distribution. Recently, Balasubramanian et al. [9] advocated the need for a sound theory for non-log-concave sampling and provided some guarantees for the unadjusted Langevin algorithm (ULA) in sampling from log-smooth (Lipschitz/Hölder smooth) densities. These results are preliminary for the ULA (and its variants) with a specific class of densities (smooth). Theoretical understandings of other classes of algorithms and densities are needed.

38th Conference on Neural Information Processing Systems (NeurIPS 2024).

We approach the subject through the lens of nonconvex optimization in the space of probability distributions and pose discretized Wasserstein gradient flows as iterative minimization algorithms. This allows us to, on the one hand, use and extend tools from classical nonconvex optimization and, on the other hand, derive more connections between sampling and optimization.

We study the following *non-geodesically-convex* optimization problem defined over the space $\mathcal{P}_2(X)$ of probability measures $\mu$ over $X = \mathbb{R}^d$ with finite second moment, i.e., $\int \|x\|^2 d\mu(x) < +\infty$,

$$\min_{\mu \in \mathcal{P}_2(X)} \mathcal{F}(\mu) := \mathcal{E}_F(\mu) + \mathscr{H}(\mu) := \mathcal{E}_{G-H}(\mu) + \mathscr{H}(\mu) \tag{1}$$

where $F : X \to \mathbb{R}$ is a *nonconvex* function which can be represented as a *difference* of two convex functions $G$ and $H$, $\mathcal{E}_F(\mu) := \int F(x)d\mu(x)$ is the potential energy, and $\mathscr{H} : \mathcal{P}_2(X) \to \mathbb{R} \cup \{+\infty\}$ plays a role as the regularizer which is assumed to be a convex function along *generalized geodesics*.

**Why difference-of-convex structure?**   Nonconvexity lies at the difference-of-convex (DC) structure $F = G - H$, where $G$ and $H$ are called the first and second DC components, respectively. $F$ being nonconvex implies $\mathcal{E}_F$ being non-geodesically-convex in general. First, the class of DC functions is very rich, and DC structures are present everywhere in real-world applications [60, 46, 47, 1, 22, 56, 58]. Weakly convex and Lipschitz smooth ($L$-smooth or simply smooth) functions are two subclasses of DC functions. Furthermore, any continuous function can be approximated by a sequence of DC functions over a compact, convex domain [8]. We also remark that many nonconvex functions admit quite natural DC decompositions, for example, an $L$-smooth function $F$ has the following splittings: $F(x) = \alpha\|x\|^2 - (\alpha\|x\|^2 - F(x))$ and $F(x) = (F(x) + \alpha\|x\|^2) - \alpha\|x\|^2$ whenever $\alpha \geq L/2$. Second, DC functions preserve enough structure to extend convex analysis. Such structure is key in classic DC programming [60] and in our Wasserstein space analysis with optimal transport tools.

**Context**   Many problems in machine learning and sampling fall into the spectrum of problem (1). For example, refer to a discussion in [65] that inspired our work. The regularizer $\mathscr{H}$ can be the internal energy [4, Sect. 10.4.3]. Under McCann's condition, the internal energy is convex along generalized geodesics [4, Prop. 9.3.9]. In particular, the negative entropy, $\mathscr{H}(\mu) = \int \log(\mu(x))d\mu(x)$ if $\mu$ is absolutely continuous w.r.t. Lebesgue measure, $+\infty$ otherwise, is a special case of internal energy satisfying McCann's condition. In the latter case, $\mathcal{F}(\mu) = \mathrm{D}_{\mathrm{KL}}(\mu \| \mu^*) + \text{const}$ where $\mu^*(x) \propto \exp(-F(x))$, the optimization problem reduces to a sampling problem with *log-DC* target distribution. In Bayesian inference, the posterior structure depends on both the prior and likelihood. If the likelihood is log-smooth, it exhibits the aforementioned DC splittings. Log-priors, often nonsmooth to capture sparsity or low rank, typically also have explicit DC structures [48, 32, 25]. In the context of infinitely wide one-layer neural networks and Maximum Mean Discrepancy [51, 7, 21], let $\mu^*$ be the optimal distribution over a network's parameters, $k$ be a given kernel, the regularizer is then the interaction energy $\mathscr{H}(\mu) = \iint k(x,y)d\mu(x)d\mu(y)$ and $F(x) = -2\int k(x,y)d\mu^*(y)$. In general, $\mathscr{H}$ is not convex along generalized geodesics and $F$ is nonconvex but not necessarily DC. When the kernel has Lipschitz gradient, we can adjust both $\mathscr{H}$ and $F$ as $\mathscr{H}(\mu) = \iint k(x,y) + \alpha\|x\|^2 + \alpha\|y\|^2 d\mu(x)d\mu(y)$ and $F(x) = -2\int k(x,y)d\mu^*(y) - 2\alpha\|x\|^2$ for some $\alpha > 0$ making $\mathscr{H}$ generalized geodesically convex and $F$ concave (hence DC); Appx. A.2.

Our idea is to minimize (1) in the space of probability distributions by discretization of the gradient flow of $\mathcal{F}$, leveraging on the JKO (Jordan, Kinderlehrer, and Otto) operator (2). In the previous work [70], this has been done with the Forward-Backward (FB) Euler discretization, but it lacks convergence analysis. Recently, Salim et al. [65] did some study on FB Euler, but their results do not apply here because $F$ is nonconvex and possibly nonsmooth. Further leveraging on the DC structure of $F$ and inspired by classical DC programming literature [60], we subtly modify the FB Euler to give rise to a scheme named semi FB Euler that enjoys major theoretical advantages as we can provide a wide range of convergence analysis. Regarding the name, "semi" addresses the splitting of the potential energy. The scheme can be nevertheless reinterpreted as FB Euler.

**Contributions**   To our knowledge, no prior work studies problem (1) when $F$ is DC. Therefore, most of the derived results in this paper are novel. We propose and analyze the semi FB Euler (4) leveraging on the classic DC optimization proof template [60, 45, 46] with a substantial accommodation of Wasserstein geometry and derive the following set of new insights:

Thm. 1 We show that if the $H$ is continuously differentiable, every cluster point of the sequence of distributions $\{\mu_n\}_{n \in \mathbb{N}}$ generated by semi FB Euler is a *critical point* to $\mathcal{F}$. Note that

criticality is a notion from the DC programming literature [60] and it is a necessary condition for local optimality; See Sect. 3.3.

**Thm. 2** We provide convergence rate of $O(N^{-1})$ in terms of Wasserstein (sub)gradient mapping in the general nonsmooth setting. The notion of gradient mapping [33, 38, 55] is from the context of proximal algorithms in Euclidean space that is applicable to nonconvex programs where the notion of distance to global solution is—in general—not possible to work out.

**Thm. 3** Under the extra assumption that $H$ is continuously twice differentiable and has bounded Hessian, we provide a convergence rate of $O(N^{-\frac{1}{2}})$ in terms of distance of 0 to the Fréchet subdifferential of $\mathcal{F}$. One can think of this as convergence rate to Fréchet stationarity, i.e., if $\mu^*$ is a Fréchet stationary point of $\mathcal{F}$, then, by definition, 0 is in the Fréchet subdifferential of $\mathcal{F}$ at $\mu^*$. Fréchet stationarity is a relatively sharp necessary condition for local optimality.

**Thm. 4, 5** Under the assumptions of Thm. 3 and additionally $\mathcal{F}$ satisfying the Łojasciewicz-type inequality for some Łojasciewicz exponent of $\theta \in [0, 1)$, we show that $\{\mu_n\}_{n \in \mathbb{N}}$ is a Cauchy sequence under Wasserstein topology, and thanks to the completeness of the Wasserstein space $\{\mu_n\}_{n \in \mathbb{N}}$ converges to some $\mu^*$. We show that $\mu^*$ is in fact a global minimizer to $\mathcal{F}$. Furthermore, we provide convergence rate of $\mu_n \to \mu^*$ in three different regimes ($W_2$ denotes the Wasserstein metric): (1) if $\theta = 0$, $W_2(\mu_n, \mu^*)$ converges to 0 after a finite number of steps; (2) if $\theta \in (0, 1/2]$, both $\mathcal{F}(\mu_n) - \mathcal{F}(\mu^*)$ and $W_2(\mu_n, \mu^*)$ converges to 0 exponentially fast; (3) if $\theta \in (1/2, 1)$, both $\mathcal{F}(\mu_n) - \mathcal{F}(\mu^*)$ and $W_2(\mu_n, \mu^*)$ converges sublinearly to 0 with rates $O\left(n^{-\frac{1}{2\theta-1}}\right)$ and $O\left(n^{-\frac{1-\theta}{2\theta-1}}\right)$, respectively. When $\mathscr{H}$ is the negative entropy, $\mathcal{F}(\mu_n) - \mathcal{F}(\mu^*) = \mathrm{D_{KL}}(\mu_n \| \mu^*)$; Therefore, in the sampling context, we provide convergence guarantees in both Wasserstein and KL distances. See Sect. 4.3 for additional observations and implications.

## 2 Preliminaries

### 2.1 Notations and basic results in measure theory and functional analysis

We denote by $X = \mathbb{R}^d$, $\mathcal{B}(X)$ the Borel $\sigma$-algebra over $X$, and $\mathscr{L}^d$ the Lebesgue measure on $X$. $\mathcal{P}(X)$ is the set of Borel probability measures on $X$. For $\mu \in \mathcal{P}(X)$, we denote its second-order moment by $\mathfrak{m}_2(\mu) := \int_X \|x\|^2 d\mu(x)$, where $\mathfrak{m}_2(\mu)$ can be infinity. $\mathcal{P}_2(X) \subset \mathcal{P}(X)$ denotes a set of finite second-order moment probability measures. $\mathcal{P}_{2,\mathrm{abs}}(X) \subset \mathcal{P}_2(X)$ is the set of measures that are absolutely continuous w.r.t. $\mathscr{L}^d$. Here $\mu$-a.e. stands for almost everywhere w.r.t. $\mu$.

Let $C^p(X), C_c^\infty(X), C_b(X)$ be the classes of $p$-time continuously differentiable functions, infinitely differentiable functions with compact support, bounded and continuous functions, respectively.

From functional analysis [20], for each $p \geq 1$, $L^p(X, \mu)$ denotes the Banach space of measurable (where measurable is understood as Borel measurable from now on) functions $f$ such that $\int_X |f(x)|^p d\mu(x) < +\infty$. We shall consider an element of $L^p(X, \mu)$ as an equivalent class of functions that agree $\mu$-a.e. on $X$ rather than a sole function. The norm of $f \in L^p(X, \mu)$ is $\|f\|_{L^p(X,\mu)} = (\int_X |f(x)|^p d\mu(x))^{1/p}$. When $p = 2$, $L^2(X, \mu)$ is actually a Hilbert space with the inner product $\langle f, g \rangle_{L^2(X,\mu)} = \int_X f(x)g(x)d\mu(x)$ which induces the mentioned norm. These results can be extended to vector-valued functions. In particular, we denote by $L^2(X, X, \mu)$ the Hilbert space of $\xi : X \to X$ in which $\|\xi\| \in L^2(X, \mu)$. The norm $\|\xi\|_{L^2(X,X,\mu)} := (\int_X \|\xi(x)\|^2 d\mu(x))^{1/2}$.

We say that $f : X \to \mathbb{R}$ has quadratic growth if there exists $a > 0$ such that $|f(x)| \leq a(\|x\|^2 + 1)$ for all $x \in X$. It is clear that if $f$ has quadratic growth and $\mu \in \mathcal{P}_2(X)$, then $f \in L^1(X, \mu)$.

The pushforward of a measure $\mu \in \mathcal{P}(X)$ through a Borel map $T : X \to \mathbb{R}^m$, denoted by $T_\#\mu$ is defined by $(T_\#\mu)(A) := \mu(T^{-1}(A))$ for every Borel sets $A \subset \mathbb{R}^m$.

### 2.2 Optimal transport [4, 5, 69, 68]

Given $\mu, \nu \in \mathcal{P}(X)$, the principal problem in optimal transport is to find a transport map $T$ pushing $\mu$ to $\nu$, i.e., $T_\#\mu = \nu$, in the most cost-efficient way, i.e., minimizing $\|x - T(x)\|^2$ on $\mu$-average. Monge's formulation for this problem is $\inf_{T:T_\#\mu=\nu} \int_X \|x - T(x)\|^2 d\mu(x)$, where the optimal

solution, if exists, is denoted by $T_\mu^\nu$ and called the optimal (Monge) map. Monge's problem can be ill-posed, e.g., no such $T_\mu^\nu$ exists when $\mu$ is a Dirac mass and $\nu$ is absolutely continuous [5].

By relaxing Monge's formulation, Kantorovich considers $\min_{\gamma \in \Gamma(\mu,\nu)} \int_{X \times X} \|x - y\|^2 d\gamma(x, y)$, where $\Gamma(\mu, \nu)$ denotes the set of probabilities over $X \times X$ whose marginals are $\mu$ and $\nu$, i.e, $\gamma \in \Gamma(\mu, \nu)$ iff $\text{proj}_{1\#}\gamma = \mu, \text{proj}_{2\#}\gamma = \nu$ where $\text{proj}_1, \text{proj}_2$ are the projections onto the first $X$ space and the second $X$ space, respectively. Such $\gamma$ is called *a plan*. Kantorovich's formulation is well-posed because $\Gamma(\mu, \nu)$ is non-empty (at least $\mu \times \nu \in \Gamma(\mu, \nu)$) and the $\arg \min$ element actually exists (see [5, Sect. 2.2]). The set of optimal plans between $\mu$ and $\nu$ is denoted by $\Gamma_o(\mu, \nu)$. In terms of random variables, any pairs $(X, Y)$ where $X \sim \mu, Y \sim \nu$ is called a coupling of $\mu$ and $\nu$ while it is called an optimal coupling if the joint law of $X$ and $Y$ is in $\Gamma_o(\mu, \nu)$.

In $\mathcal{P}_2(X)$, the $\min$ value in Kantorovich's problem specifies a *valid* metric referred to as Wasserstein distance, $W_2(\mu, \nu) = (\int_{X \times X} \|x - y\|^2 d\gamma(x, y))^{1/2}$ for some, and thus all, $\gamma \in \Gamma_o(\mu, \nu)$. The metric space $(\mathcal{P}_2(X), W_2)$ is then called the Wasserstein space. In $\mathcal{P}_2(X)$, beside the convergence notion induced by the Wasserstein metric, there is a weaker notion of convergence called *narrow convergence*: we say a sequence $\{\mu_n\}_{n \in \mathbb{N}} \subset \mathcal{P}_2(X)$ converges narrowly to $\mu \in \mathcal{P}_2(X)$ if $\int_X \phi(x) d\mu_n(x) \to \int_X \phi(x) d\mu(x)$ for all $\phi \in C_b(X)$. Convergence in the Wasserstein metric implies narrow convergence but the converse is not necessarily true. The extra condition to make it true is $\mathfrak{m}_2(\mu_n) \to \mathfrak{m}_2(\mu)$. We denote Wasserstein and narrow convergence by $\xrightarrow{\text{Wass}}$ and $\xrightarrow{\text{narrow}}$, respectively.

If $\mu \in \mathcal{P}_{2,\text{abs}}(X), \nu \in \mathcal{P}_2(X)$, Monge's formulation is well-posed and the unique ($\mu$-a.e.) solution exists, and in this case, it is safe to talk about (and use) the optimal transport map $T_\mu^\nu$. Moreover, there exists some convex function $f$ such that $T_\mu^\nu = \nabla f$ $\mu$-a.e. Kantorovich's problem also has a unique solution $\gamma$ and it is given by $\gamma = (I, T_\mu^\nu)_\#\mu$ where $I$ is the identity map. This is known as Brenier theorem or polar factorization theorem [18].

## 2.3 Subdifferential calculus in the Wasserstein space

Apart from being a metric space, $(\mathcal{P}_2(X), W_2)$ also enjoys some pre-Riemannian structure making subdifferential calculus on it possible. Let us have a picture of a *manifold* in mind. Firstly, the tangent space [4] of $\mathcal{P}_2(X)$ at $\mu$ is $\text{Tan}_\mu \mathcal{P}_2(X) := \overline{\{\nabla \psi : \psi \in C_c^\infty(X)\}}^{L^2(X,X,\mu)}$, where the closure is w.r.t. the $L^2(X, X, \mu)$-topology. Intuitively, for $\psi \in C_c^\infty(X)$, $I + \epsilon \nabla \psi$ is an optimal transport map if $\epsilon > 0$ is small enough [43], so $\nabla \psi$ plays a role as "tangent vector".

Let $\phi : \mathcal{P}_2(X) \to \mathbb{R} \cup \{+\infty\}$, we denote $\text{dom}(\phi) = \{\mu \in \mathcal{P}_2(X) : \phi(\mu) < +\infty\}$. Let $\mu \in \text{dom}(\phi)$, we say that a map $\xi \in L^2(X, X, \mu)$ belongs to the *Fréchet subdifferential* [15, 43] $\partial_F^-\phi(\mu)$ if $\phi(\nu) - \phi(\mu) \geq \sup_{\gamma \in \Gamma_o(\mu,\nu)} \int_{X \times X} \langle \xi(x), y - x \rangle d\gamma(x, y) + o(W_2(\mu, \nu))$ for all $\nu \in \mathcal{P}_2(X)$, where the little-o notation means $\lim_{s \to 0} o(s)/s = 0$. If $\partial_F^-\phi(\mu) \neq \emptyset$, we say $\phi$ is Fréchet subdifferentiable at $\mu$. We also denote $\text{dom}(\partial_F^-\phi) = \{\mu \in \mathcal{P}_2(X) : \partial_F^-\phi(\mu) \neq \emptyset\}$.

Similarly, we say that $\xi \in L^2(X, X, \mu)$ belongs to the (Fréchet) superdifferential $\partial_F^+\phi(\mu)$ of $\phi$ at $\mu$ if $-\xi \in \partial_F^-(-\phi)(\mu)$. In other words, $\partial_F^-(-\phi)(\mu) = -\partial_F^+\phi(\mu)$.

We say $\phi$ is Wasserstein differentiable [15, 43] at $\mu \in \text{dom}(\phi)$ if $\partial_F^-\phi(\mu) \cap \partial_F^+\phi(\mu) \neq \emptyset$. We call an element of the intersection, denoted by $\nabla_W \phi(\mu)$, a Wasserstein gradient of $\phi$ at $\mu$, and it holds $\phi(\nu) - \phi(\mu) = \int_{X \times X} \langle \nabla_W \phi(\mu)(x), y - x \rangle d\gamma(x, y) + o(W_2(\mu, \nu))$, for all $\nu \in \mathcal{P}_2(X)$ and any $\gamma \in \Gamma_o(\mu, \nu)$. The Wasserstein gradient is not unique in general, but its parallel component in $\text{Tan}_\mu \mathcal{P}_2(X)$ is unique, and this parallel component is again a valid Wasserstein gradient as the orthogonal component plays no role in the above definitions, i.e., if $\xi^\perp \in \text{Tan}_\mu \mathcal{P}_2(X)^\perp$, it holds $\int_{X \times X} \langle \xi^\perp(x), y - x \rangle d\gamma(x, y) = 0$ for any $\nu \in \mathcal{P}_2(X)$ and $\gamma \in \Gamma_o(\mu, \nu)$ [43, Prop. 2.5]. We may refer to this parallel component as the *unique* Wasserstein gradient of $\phi$ at $\mu$.

## 2.4 Optimization in the Wasserstein space

A function $\phi : \mathcal{P}_2(X) \to \mathbb{R} \cup \{+\infty\}$ is called *proper* if $\text{dom}(\phi) \neq \emptyset$, while it is called lower semicontinuous (l.s.c) if for any sequence $\mu_n \xrightarrow{\text{Wass}} \mu$, it holds $\liminf_n \phi(\mu_n) \geq \phi(\mu)$.

We next recall (a simplified version of) *generalized* geodesic convexity.

**Definition 1.** *[65] Let $\phi : \mathcal{P}_2(X) \to \mathbb{R} \cup \{+\infty\}$. We say $\phi$ is convex along generalized geodesics if $\forall \mu, \pi \in \mathcal{P}_2(X), \forall \nu \in \mathcal{P}_{2,\mathrm{abs}}(X), \phi((tT_\nu^\mu + (1-t)T_\nu^\pi)_\#\nu) \leq t\phi(\mu) + (1-t)\phi(\pi), \forall t \in [0,1].$*

The curve $t \mapsto (tT_\nu^\mu + (1-t)T_\nu^\pi)_\#\nu$ (called a generalized geodesic) interpolates from $\pi$ to $\mu$ as $t$ runs from 0 to 1. The definition says that $\phi$ is convex along these curves. If $\mu \in \mathcal{P}_{2,\mathrm{abs}}(X)$ and $\nu = \mu$, the curve is a geodesic in $(\mathcal{P}_2(X), W_2)$. If the definition is relaxed to the class of geodesics only, we say that $\phi$ is convex along geodesics.

An important characterization of Fréchet subdifferential of a geodesically convex function is that we can drop the little-o notation in its definition in Sect. 2.3 [4, Sect 10.1.1]. As a convention, for a geodesically convex function $\phi$, the Fréchet subdifferential $\partial_F^-$ will be simply written as $\partial$.

**First-order optimality conditions**    Let $\phi : \mathcal{P}_2(X) \to \mathbb{R} \cup \{+\infty\}$ be a proper function. $\mu^* \in \mathcal{P}_2(X)$ is a global minimizer of $\phi$ if $\phi(\mu^*) \leq \phi(\mu), \forall \mu \in \mathcal{P}_2(X)$. For local optimality, we shall use the Wasserstein metric to define neighborhoods. $\mu^* \in \mathcal{P}_2(X)$ is a local minimizer if there exists $r > 0$ such that $\phi(\mu^*) \leq \phi(\mu)$ for all $\mu : W_2(\mu, \mu^*) < r$. We shall denote $B(\mu^*, r) := \{\mu \in \mathcal{P}_2(X) : W_2(\mu, \mu^*) < r\}$ the (open) Wasserstein ball centered at $\mu^*$ with radius $r$. If we replace $<$ by $\leq$ we obtain the notion of a closed Wasserstein ball.

We call $\mu^*$ a Fréchet stationary point of $\phi$ if $0 \in \partial_F^- \phi(\mu^*)$. Fréchet stationarity is a necessary condition for local optimality. In other words, if $\mu^*$ is a local minimizer, it is a Fréchet stationary point (Lem. 5 in Appendix). In addition, if $\phi$ is Wasserstein differentiable at $\mu^*$, $\nabla_W \phi(\mu^*)(x) = 0$ $\mu^*$-a.e. [43]. When $\phi$ is geodesically convex, Fréchet stationarity is a sufficient condition for global optimality (Lem. 6 in Appendix).

# 3   Semi Forward-Backward Euler for difference-of-convex structures

## 3.1   Wasserstein gradient flows: different types of discretizations

To neatly present the idea of minimizing $\mathcal{F}$ via discretized gradient flow, we first assume for a moment that $F$ is infinitely differentiable and $\mathcal{H}$ is the negative entropy. See also a discussion in [65].

We wish to minimize (1) in the space of probability distributions. A natural idea is to apply discretizations of the gradient flow of $\mathcal{F}$, where the gradient flow is defined (under some technical assumptions [39]) as the limit $\eta \to 0^+$ of the following scheme with some simple time-interpolation

$$\mu_{n+1} \in \mathrm{JKO}_{\eta\mathcal{F}}(\mu_n), \text{ where } \mathrm{JKO}_{\eta\mathcal{F}}(\mu) := \operatorname*{arg\,min}_{\nu \in \mathcal{P}_2(X)} \mathcal{F}(\nu) + \frac{1}{2\eta}W_2^2(\mu, \nu). \qquad (2)$$

Straightforwardly, given a fixed $\eta > 0$, (2) gives back a discretization for this flow known as Backward Euler. On the other hand, if $\mathcal{F}$ is Wasserstein differentiable (Sect. 2.2), the Forward Euler discretization reads [70] $\mu_{n+1} = (I - \eta\nabla_W\mathcal{F}(\mu_n))_\#\mu_n$, which is reinterpreted as doing gradient descent in the space of probability distributions. These are optimization methods that work *directly* on the objective function $\mathcal{F}$ itself. However, the composite structure of $\mathcal{F}$ (a sum of several terms) can also be exploited. One such scheme is the unadjusted Langevin algorithm (ULA), where it first takes a gradient step w.r.t. the potential part, then follows the heat flow corresponding to the entropy part [70]: $\nu_{n+1} = (I - \eta\nabla F)_\#\mu_n$, and $\mu_{n+1} = \mathcal{N}(0, 2\eta I) * \nu_{n+1}$, where $*$ is the convolution. This ULA is "viewed" in the space of distributions (Eulerian approach), a more familiar and equivalent form of the ULA from the particle perspective (Lagrangian approach) goes like $x_{n+1} = x_n - \eta\nabla F(x_n) + \sqrt{2\eta}z_k$ where $z_k \sim \mathcal{N}(0, I)$. The ULA is known to be asymptotically biased even for Gaussian target measure (Ornstein-Uhlenbeck process). To correct this bias, the Metropolis-Hasting accept-reject step [62] is sometimes introduced. Metropolis-Hasting algorithm [52, 36] is a much more general framework that works with quite any proposal (e.g., a random walk) whose convergence analysis is based on the Markov kernel satisfying the detailed balance condition. This convergence framework is different from what is considered in this work: we are more interested in the underlying dynamics of the chain. Metropolis-Hasting algorithm is indeed another story.

In optimization, for composite structure, Forward-Backward (FB) Euler and its variants are methods of choice [59, 10]. The corresponding FB Euler for $\mathcal{F}$ will take the gradient step (forward) according to the potential, and JKO step (backward) w.r.t. the negative entropy

$$(\text{FB Euler}) \quad \nu_{n+1} = (I - \eta\nabla F)_\#\mu_n, \text{ and } \mu_{n+1} \in \mathrm{JKO}_{\eta\mathcal{H}}(\nu_{n+1}). \qquad (3)$$

This scheme appears in [70] without convergence analysis, and later on [65] derives non-asymptotic convergence guarantees under the assumption $F$ being convex and Lipschitz smooth.

In this work, as $F$ is nonconvex and nonsmooth, the theory in [65] does not apply, and the convergence (if any) of (3) remains mysterious. The DC structure of $F$ can be further exploited. In DC programming [60], the forward step should be applied to the concave part, while the backward step should be applied to the convex part. We hence propose the following semi FB Euler

$$\text{(semi FB Euler)} \quad \nu_{n+1} = (I + \eta \nabla H)_{\#}\mu_n, \text{ and } \mu_{n+1} \in \text{JKO}_{\eta(\mathscr{H}+\mathcal{E}_G)}(\nu_{n+1}) \tag{4}$$

for which we can provide convergence guarantees. Apparently, the difference between semi FB Euler and FB Euler is subtle: while FB Euler does forward on $\mathcal{E}_{G-H} = \mathcal{E}_G - \mathcal{E}_H$ and backward on $\mathscr{H}$, semi FB Euler does forward on $-\mathcal{E}_H$ and backward on $\mathscr{H} + \mathcal{E}_G$; recall that $\mathcal{F} = \mathcal{E}_G - \mathcal{E}_H + \mathscr{H}$.

Theoretically, semi FB Euler enjoys some advantages compared to FB Euler. Thanks to Brenier theorem (Sect. 2.2), the pushing step in semi FB Euler is *optimal* since $H$ is convex; Meanwhile, the pushing in FB Euler is non-optimal whose optimal Monge map is not identifiable in general. The convergence of FB Euler is still an open question, even when $F$ is (DC) differentiable. In contrast, we can provide a solid theoretical guarantee for semi FB Euler, especially when $H$ is differentiable. Additionally, we also offer convergence guarantees when $H$ is nonsmooth.

## 3.2 Problem setting

Our goal is to minimize the non-geodesically-convex functional $\mathcal{F}(\mu) = \mathcal{E}_F(\mu) + \mathscr{H}(\mu)$ over $\mathcal{P}_2(X)$, where $F = G - H$ is a DC function. We make Assumption 1 throughout the paper:

**Assumption 1.**     *(i) The objective function $\mathcal{F}$ is bounded below.*

    *(ii) $G, H : X \to \mathbb{R}$ are convex functions and have quadratic growth.*

    *(iii) $\mathscr{H} : \mathcal{P}_2(X) \to \mathbb{R} \cup \{+\infty\}$ is proper, l.s.c, and convex along generalized geodesics in $(\mathcal{P}_2(X), W_2)$, and $\text{dom}(\mathscr{H}) \subset \mathcal{P}_{2,\text{abs}}(X)$.*

    *(iv) There exists $\eta_0 > 0$ such that $\forall \eta \in (0, \eta_0)$, $\text{JKO}_{\eta(\mathcal{E}_G+\mathscr{H})}(\mu) \neq \emptyset$ for every $\mu \in \mathcal{P}_2(X)$.*

Note that Assumption 1(iv) is a commonly-used assumption to simplify technical complication when working with the JKO operator [4, 15, 65]. Assumption 1(ii) implies $\mathcal{E}_G$ and $\mathcal{E}_H$ are continuous w.r.t. Wasserstein topology [3, Prop. 2.4] ($G, H$ are continuous [54, Cor. 2.27] and have quadratic growth).

## 3.3 Optimality charactizations

First, it follows from Assumption 1(iii), $\text{dom}(\mathcal{F}) \subset \mathcal{P}_{2,\text{abs}}(X)$. By analogy to DC programming in Euclidean space, we call $\mu^* \in \text{dom}(\mathcal{F})$ a *critical point* of $\mathcal{F} = \mathscr{H} + \mathcal{E}_G - \mathcal{E}_H$ if $\partial(\mathscr{H} + \mathcal{E}_G)(\mu^*) \cap \partial \mathcal{E}_H(\mu^*) \neq \emptyset$. Criticality is a necessary condition for local optimality (Lem. 7). Moreover, if $\mathcal{E}_H$ is Wasserstein differentiable at $\mu^*$, criticality becomes Fréchet stationarity (Lem. 8).

## 3.4 Semi FB Euler: a general setting

We allow $H$ to be non-differentiable in some derivations, meaning that $\partial H$ (convex subdifferential [54]) contains multiple elements in general. We first pick a selector $S$ of $\partial H$, i.e., $S : X \to X$, such that $S(x) \in \partial H(x)$. By the axiom of choice (Zermelo, 1904, see, e.g., [37]), such selection always exists. However, an arbitrary selector can behave badly, e.g., not measurable. We shall first restrict ourselves to the class of Borel measurable selectors (see Appx. A.1 for an existence discussion).

**Assumption 2** (Measurability). *The selector $S$ is Borel measurable.*

We recall the semi FB scheme (4) but for nonsmooth $F$ as follows: start with an initial distribution $\mu_0 \in \mathcal{P}_{2,\text{abs}}(X)$, given a discretization stepsize $0 < \eta < \eta_0$, we repeat the following two steps:

$$\nu_{n+1} = (I + \eta S)_{\#}\mu_n \quad \triangleleft \text{ push forward step}; \quad \mu_{n+1} = \text{JKO}_{\eta(\mathcal{E}_G+\mathscr{H})}(\nu_{n+1}) \quad \triangleleft \text{ JKO step}.$$

Well-definiteness and properties: Given $\mu_n \in \mathcal{P}_2(X)$, it follows from Lem. (4) that $\nu_{n+1} \in \mathcal{P}_2(X)$. The two generated sequences are then in $\mathcal{P}_2(X)$. Moreover, it follows from Assumption 1 that $\{\mu_n\}_{n\in\mathbb{N}}$ are in $\mathcal{P}_{2,\text{abs}}(X)$, so are $\{\nu_n\}_{n\in\mathbb{N}}$ using Lem. 9 by noting that $I + \eta S$ is subgradient of a strongly convex function $x \mapsto (1/2)\|x\|^2 + \eta H(x)$.

# 4 Convergence analysis

## 4.1 Asymptotic analysis

**Lemma 1** (Descent lemma). *Under Assumptions 1 and 2, let $\{\mu_n\}_{n\in\mathbb{N}}$ be the sequence of distributions produced by semi FB Euler starting from some $\mu_0 \in \mathcal{P}_{2,\mathrm{abs}}(X)$ with $0 < \eta < \eta_0$. Then it holds $\mathcal{F}(\mu_{n+1}) \leq \mathcal{F}(\mu_n) - \frac{1}{\eta}\int_X \|T^{\mu_n}_{\nu_{n+1}}(x) - T^{\mu_{n+1}}_{\nu_{n+1}}(x)\|^2 d\nu_{n+1}(x), \quad \forall n \in \mathbb{N}.$*

Lem. 1 shows that the objective does not increase along semi FB Euler's iterates. Proof of Lem. 1 is in Appx. A.3. By using Lem. 1, we establish asymptotic convergence for semi FB Euler as follows.

For the asymptotic convergence analysis, we need the following assumption on $H$.

**Assumption 3.** *$H$ is continuously differentiable.*

**Theorem 1** (Asymptotic convergence). *Under Assumptions 1, 3, let $\{\mu_n\}_{n\in\mathbb{N}}$ and $\{\nu_n\}_{n\in\mathbb{N}}$ be sequences produced by semi FB Euler starting from some $\mu_0 \in \mathcal{P}_{2,\mathrm{abs}}(X)$ with $0 < \eta < \eta_0$. If $\{\mu_n\}_{n\in\mathbb{N}}$ is relatively compact with respect to the Wasserstein topology and $\sup_{n\in\mathbb{N}} \mathscr{H}(\nu_n) < +\infty$, then every cluster point of $\{\mu_n\}_{n\in\mathbb{N}}$ is a critical point of $\mathcal{F}$.*

Proof of Thm.1 is in Appx. A.4. Thm. 1 does not ensure convergence of the whole sequence $\{\mu_n\}_{n\in\mathbb{N}}$; Rather, it guarantees subsequential convergence to critical points of $\mathcal{F}$.

**Remark 1.** *In the Euclidean space, the compactness assumption of the generated sequence is usually enforced via the* coercivity *assumption: $f(x) \to +\infty$ whenever $\|x\| \to +\infty$. A striking difference in the Wasserstein space is that closed Wasserstein balls are not compact in the Wasserstein topology [43, Prop. 4.2], making coercivity not sufficient to induce (Wasserstein) compactness. For Thm. 1, we simply assume the sequence $\{\mu_n\}_{n\in\mathbb{N}}$ to be relatively compact.*

## 4.2 Non asymptotic analysis

To measure how fast the algorithm converges, we need some convergence measurement. First, for proximal-type algorithms in Euclidean space, the notion of *gradient mapping* $\mathcal{G}_\eta(x_n)$ is usually used (see, e.g., [33, 55] and [38, Eq. (5)]) and we measure the rate $\|\mathcal{G}_\eta(x_n)\|^2 \to 0$. In analogy as in Euclidean space, we define the *Wasserstein (sub)gradient mapping* as follows $\mathcal{G}_\eta(\mu) := \frac{1}{\eta}\left(I - T^{\mathrm{JKO}_{\eta(\mathcal{E}_G + \mathscr{H})}((I + \eta S)_\# \mu)}_\mu\right)$, and we measure the rate of $\|\mathcal{G}_\eta(\mu_n)\|^2_{L^2(X,X,\mu_n)} \to 0$.

**Theorem 2** (Convergence rate: Wasserstein (sub)gradient mapping). *Under Assumptions 1, 2, let $\{\mu_n\}_{n\in\mathbb{N}}$ be the sequence of distributions produced by semi FB Euler starting from some $\mu_0 \in \mathcal{P}_{2,\mathrm{abs}}(X)$ with $0 < \eta < \eta_0$. Then it holds $\min_{n=\overline{1,N}} \|\mathcal{G}_\eta(\mu_n)\|^2_{L^2(X,X,\mu_n)} = O(N^{-1})$.*

Proof of Thm. 2 is in Appx. A.5. This theorem holds without requiring $G$ and $H$ to be differentiable.

Next, if $H$ is twice differentiable with uniformly bounded Hessian, we can derive a stronger convergence guarantee based on Fréchet stationarity (see Sect. 2.4). In other words, we evaluate the rate of $\mathrm{dist}\left(0, \partial_F^- \mathcal{F}(\mu_n)\right) := \inf_{\xi \in \partial_F^- \mathcal{F}(\mu_n)} \|\xi\|_{L^2(X,X;\mu_n)} \to 0$.

**Assumption 4.** *$H \in C^2(X)$ whose Hessian is bounded uniformly ($H$ is then $L_H$-smooth).*

**Theorem 3** (Convergence rate: Fréchet subdifferentials). *Under Assumptions 1, 4, let $\{\mu_n\}_{n\in\mathbb{N}}$ be the sequence of distributions produced by semi FB Euler starting from some $\mu_0 \in \mathcal{P}_{2,\mathrm{abs}}(X)$ with $0 < \eta < \eta_0$, then $\min_{n=\overline{1,N}} \mathrm{dist}\left(0, \partial_F^- \mathcal{F}(\mu_n)\right) = O\left(N^{-\frac{1}{2}}\right).$*

Proof of Thm. 3 is in Appx. A.6.

## 4.3 Fast convergence under isoperimetry and beyond

Fast convergence can be obtained under *isoperimetry*, e.g., log-Sobolev inequality (LSI). There are certain connections between LSI in sampling and the Łojasiewicz condition in optimization allowing linear convergence. In nonconvex optimization in Euclidean space, analytic and subanalytic functions are a large class satisfying Łojasiewicz condition [44, 14]. Subanalytic DC programs are studied in [45]. In the infinite-dimensional setting of the Wasserstein space, the Łojasiewicz condition should be regarded as functional inequalities [12].

**Assumption 5** (Łojasiewicz condition in the Wasserstein space). *Assume that $\mathcal{F}^*$ is the optimal value of $\mathcal{F}$, and assume there exist $r_0 \in (\mathcal{F}^*, +\infty]$, $\theta \in [0,1)$, and $c > 0$ such that for all $\mu \in \mathcal{P}_2(X)$, $\mathcal{F}(\mu) - \mathcal{F}^* < r_0 \Rightarrow c\,(\mathcal{F}(\mu) - \mathcal{F}^*)^\theta \leq \inf\{\|\xi\|_{L^2(X,X,\mu)} : \xi \in \partial_F^- \mathcal{F}(\mu)\}$, where the conventions $0^0 = 0$ and $\inf \emptyset = +\infty$ are used. We call $\theta \in [0,1)$ the Łojasiewicz exponent of $\mathcal{F}$ at optimality.*

**Remark 2.** *If $\mathscr{H}$ is the is negative entropy, $F \in C^2(X)$ whose Hessian is bounded uniformly, then $\mathcal{F}$ is Wasserstein differentiable at $\mu \in \mathcal{P}_{2,\mathrm{abs}}(X)$ with gradient $\nabla_W \mathcal{F}(\mu) = \frac{\nabla \mu}{\mu} + \nabla F$ provided that all terms are well-defined [43, Prop. 2.12, E.g. 2.3]. We have $\|\nabla_W \mathcal{F}(\mu)\|_{L^2(X,X,\mu)}^2 = \int \left\| \frac{\nabla \mu(x)}{\mu(x)} + \nabla F(x) \right\|^2 d\mu(x) = \int \mu(x) \left\| \nabla \log \frac{\mu(x)}{\mu^*(x)} \right\|^2 dx$, where $\mu^* \propto \exp(-F)$. On the other hand, $\mathcal{F}(\mu) - \mathcal{F}^* = \mathrm{D}_{\mathrm{KL}}(\mu \| \mu^*)$. The log-Sobolev inequality with parameter $\alpha > 0$ inequality reads [57] $\mathrm{D}_{\mathrm{KL}}(\mu \| \mu^*) \leq \frac{1}{2\alpha} \mathrm{FI}(\mu \| \mu^*) := \frac{1}{2\alpha} \int \mu(x) \left\| \nabla \log \frac{\mu(x)}{\mu^*(x)} \right\|^2 dx$, where $\mathrm{FI}(\mu \| \mu^*)$ is the relative Fisher information of $\mu$ w.r.t. $\mu^*$. Therefore, log-Sobolev inequality is a special case of Łojasiewicz condition with $\theta = 1/2$. In another case, when the objective function is the Maximum Mean Discrepancy, under some regularity assumption of the kernel, it holds [7]*

$$2(\mathcal{F}(\mu) - \mathcal{F}(\mu^*)) \leq \|\mu^* - \mu\|_{\dot{H}^{-1}(\mu)} \times \int \|\nabla_W \mathcal{F}(\mu)\|^2 d\mu(x)$$

*where $\|\mu^* - \mu\|_{\dot{H}^{-1}(\mu)}$ is the weighted negative Sobolev distance. This is "nearly" the Łojasiewicz condition, with a caveat that $\|\mu^* - \mu\|_{\dot{H}^{-1}(\mu)}$ may be unbounded. Nevertheless, assuming the boundedness of this term along the algorithm's iterates is sufficient for convergence.*

**Theorem 4.** *Under Assumptions 1, 4 and Assumption 5 with parameters $(r_0, c, \theta)$. Let $\{\mu_n\}_{n \in \mathbb{N}}$ be the sequence of distributions produced by semi FB Euler starting from some sufficiently warm-up $\mu_0 \in \mathcal{P}_{2,\mathrm{abs}}(X)$ such that $\mathcal{F}(\mu_0) < r_0$ and with stepsize $0 < \eta < \eta_0$, then*

*(i) if $\theta = 0$, $\mathcal{F}(\mu_n) - \mathcal{F}^*$ converges to 0 in a finite number of steps;*

*(ii) if $\theta \in (0, 1/2]$, $\mathcal{F}(\mu_n) - \mathcal{F}^* = O\left(\left(\frac{M}{M+1}\right)^n\right)$ where $M = \frac{2(\eta^2 L_H^2 + 1)}{c^2 \eta}$;*

*(iii) if $\theta \in (1/2, 1)$, $\mathcal{F}(\mu_n) - \mathcal{F}^*$ converges sublinearly to 0, i.e., $\mathcal{F}(\mu_n) - \mathcal{F}^* = O\left(n^{-\frac{1}{2\theta - 1}}\right)$.*

Proof of Thm. 4 is in Appx. A.7.

**Remark 3.** *In the usual sampling case, i.e., $\mathscr{H}$ is the negative entropy, and under log-Sobolev condition, $r_0 = +\infty$. Therefore, $\mu_0$ can be arbitrarily in $\mathcal{P}_{2,\mathrm{abs}}(X)$. In the general case, however, a good enough starting point (i.e., $\mathcal{F}(\mu_0) < r_0$) is needed to guarantee we are in the region where Łojasiewicz condition comes into play. In such a case, $\mathcal{F}(\mu_n) - \mathcal{F}^* = \mathrm{D}_{\mathrm{KL}}(\mu_n \| \mu^*)$ where $\mu^*(x) \propto \exp(-F(x))$ is the target distribution (see Rmk. 2), so Thm. 4 provides convergence rate of $\{\mu_n\}_{n \in \mathbb{N}}$ to $\mu^*$ in terms of KL divergence and this convergence is exponentially fast if $\theta \in (0, 1/2]$.*

**Theorem 5.** *Under the same set of assumptions as in Thm. 4, the sequence $\{\mu_n\}_{n \in \mathbb{N}}$ is a Cauchy sequence under Wasserstein topology. Furthermore, as the Wasserstein space $(\mathcal{P}_2(X), W_2)$ is complete [5, Thm. 2.2], every Cauchy sequence is convergent, i.e., there exists $\mu^* \in \mathcal{P}_2(X)$ such that $\mu_n \xrightarrow{Wass} \mu^*$. The limit distribution $\mu^*$ is indeed the global minimizer of $\mathcal{F}$. In addition:*

*(i) if $\theta = 0$, $W_2(\mu_n, \mu^*)$ converges to 0 in a finite number of steps;*

*(ii) if $\theta \in (0, 1/2]$, $W_2(\mu_n, \mu^*) = O\left(\left(\frac{M}{M+1}\right)^n\right)$, where $M = 1 + \frac{(2(\eta^2 L_H^2 + 1))^{\frac{1}{2\theta}}}{(1-\theta)\eta^{\frac{1-\theta}{\theta}} c^{\frac{1}{\theta}}}$;*

*(iii) if $\theta \in (1/2, 1)$, $W_2(\mu_n, \mu^*) = O\left(n^{-\frac{1-\theta}{2\theta - 1}}\right)$.*

Proof of Thm. 5 is in Appx. A.8. This theorem provides convergence to optimality in terms of Wasserstein distance.

**Remark 4.** *If $\mathscr{H}$ is the negative entropy and $\theta = 1/2$, under some technical assumptions on $F$ (e.g., continuously twice differentiable), LSI implies Talagrand inequality [57] (in optimization, known as Łojasiewicz implies quadratic growth [40]), meaning that KL divergence controls squared Wasserstein distance, so fast convergence under KL divergence implies fast convergence under Wasserstein distance.*

# 5 Practical implementations

The push-forward step $\nu_{n+1} = (I + \eta \nabla H)_{\#} \mu_n$ is rather straightforward: if $Z$ are samples from $\mu_n$ then $Z + \eta \nabla H(Z)$ are samples from $\nu_{n+1}$. On the other hand, to move from $\nu_{n+1}$ to $\mu_{n+1}$ we have to work out the JKO operator. Recent advances [53, 2] propose using the gradient of an input-convex neural network (ICNN) [6] to approximate the optimal Monge map pushing $\nu_{n+1}$ to $\mu_{n+1}$, which we briefly describe as follows. This approach is inspired by Brenier theorem asserting that an optimal Monge map has to be the (sub)gradient field of some convex function. Therefore, one can "parametrize" $\mu \in \mathcal{P}_{2,\mathrm{abs}}(X)$ as $\mu = \nabla \psi_{\#} \nu_{n+1}$ for some convex function $\psi$. We then write the JKO objective as

$$\mathcal{H}(\nabla \psi_{\#} \nu_{n+1}) + \int_X G(\nabla \psi(x)) d\nu_{n+1}(x) + \frac{1}{2\eta} \int_X \|x - \nabla \psi(x)\|^2 d\nu_{n+1}(x). \tag{5}$$

While the two last terms (potential energy and squared Wasserstein distance) in (5) can be handled efficiently by the Monte Carlo method using samples from $\nu_{n+1}$, the first term $\mathcal{H}$ might be complicated as it possibly involves the (unavailable) density of $\nu_{n+1}$. We remark that the easy case would be $\mathcal{H}$ being another potential energy or an interaction energy. In such a case, Monte Carlo approximations are again readily applicable. The tricky case would be $\mathcal{H}$ being the negative entropy that requires the density of $\nu_{n+1}$. Fortunately, we have the following change of entropy formula: for any $T : X \to X$ diffeomorphic, any $\rho \in \mathcal{P}_{2,\mathrm{abs}}(X)$, it holds $-\mathcal{H}(T_{\#}\rho) = -\mathcal{H}(\rho) + \int_X \log |\det \nabla T(x)| d\rho(x)$. Therefore, (5) can be written as (up to a constant that does not depend on $\psi$)

$$\int_X \left[ -\log \det \nabla^2 \psi(x) + G(\nabla \psi(x)) + \frac{1}{2\eta} \|x - \nabla \psi(x)\|^2 \right] d\nu_{n+1}(x).$$

Note that this entropy formula can be extended naturally to the case of general internal energy [2], which means we can also handle this general case. Let us now consider the entropy case for simplicity. We can leverage on a class of input convex neural networks [6] $\psi_\theta(x)$ ($\theta$ is the neural network's parameters, $x$ is the input) in which $x \mapsto \psi_\theta(x)$ is convex. Optimizing over $\theta$ can then be solved effectively by standard deep learning optimizers (e.g. Adam). The complete scheme is given in Alg. 1. We also remark that [2] further proposes fast approximation for $\log \det \nabla^2 \psi$ and [31] leverages on the variational formula of the KL to propose an even faster scheme. Nevertheless, these schemes are generally expensive. For illustrative purposes, we adopt the vanilla version of [53]. The iteration complexity is thus cubic in $d$, linear in the size of the ICNN, and linear in the iteration count $k$ [53].

---

**Algorithm 1** Semi FB Euler for sampling

---

**Input:** Initial measure $\mu_0 \in \mathcal{P}_{2,\mathrm{abs}}(X)$, discretization step size $\eta > 0$, number of steps $K > 0$, batch size $B$.
**for** $k = 1$ to $K$ **do**
    **for** $i = 1, 2, \ldots$ **do**
        Draw a batch of samples $Z \sim \mu_0$ of size $B$;
        $\Xi \leftarrow (I + \eta \nabla H) \circ \nabla_x \psi_{\theta_k} \circ (I + \eta \nabla H) \circ \nabla_x \psi_{\theta_{k-1}} \circ \ldots \circ (I + \eta \nabla H)(Z)$;
        $\widehat{W_2^2} \leftarrow \frac{1}{B} \sum_{\xi \in \Xi} \|\nabla_x \psi_\theta(\xi) - \xi\|^2$;
        $\widehat{\mathcal{U}} \leftarrow \frac{1}{B} \sum_{\xi \in \Xi} G(\nabla_x \psi_\theta(\xi))$;
        $\widehat{\Delta \mathcal{H}} \leftarrow -\frac{1}{B} \sum_{\xi \in \Xi} \log \det \nabla_x^2 \psi_\theta(\xi)$.
        $\widehat{\mathcal{L}} \leftarrow \frac{1}{2\eta} \widehat{W_2^2} + \widehat{\mathcal{U}} + \widehat{\Delta \mathcal{H}}$.
        Apply an optimization step (e.g., Adam) over $\theta$ using $\nabla_\theta \widehat{\mathcal{L}}$.
    **end for**
    $\theta_{k+1} \leftarrow \theta$.
**end for**

---

# 6 Numerical illustrations

We perform numerical sampling experiments from non-log-concave distributions: the Gaussian mixture distribution and the distance-to-set-prior [61] relaxed von Mises–Fisher distribution. Both

are log-DC and the latter has *non-differentiable* logarithmic probability density (see Appx. C). Fig. 1 presents the sampling results. Experiment details are in Appx. B and Appx. C[1].

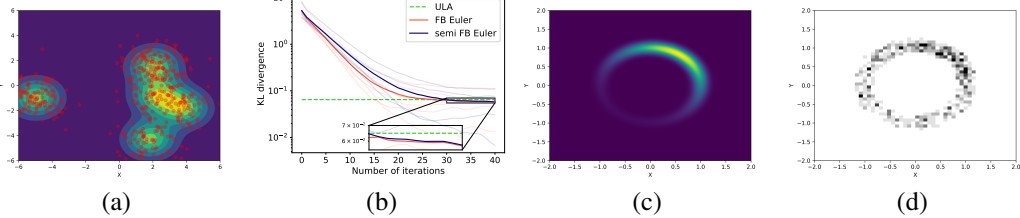

(a)  (b)  (c)  (d)

Figure 1: **(a) and (b)**: Mixture of Gaussians. **(a)** shows samples obtained from semi FB Euler at iteration 40 and **(b)** shows KL divergence along the training process: semi FB Euler with sound theory is as fast as FB Euler. We also show the ULA's final result as a horizontal line for reference; **(c) and (d)**: Relaxed von Mises-Fisher. **(c)** shows true probability density, and **(d)** shows the sample histogram obtained from semi FB Euler. In this experiment, FB Euler fails to work, attributed to the high curvature of the relaxed von Mises-Fisher.

## 7 Discussion and related work

We first narrow down our discussion on FB Euler and its variants in the Wasserstein space. When $\mathcal{H}$ is the negative entropy, Wibisono [70] provides some insightful discussion on how FB Euler should be consistent (no asymptotic bias) because the backward step is adjoint to the forward step, hence preserves stationarity. However, no convergence theory is presented for FB Euler in the Wasserstein space in [70]. Recently, Salim et al. [65] provide convergence guarantee for FB Euler within the following setting: $\mathcal{H}$ is convex along generalized geodesics, $F$ is Lipschitz smooth and convex/strongly convex. This setting remains a "convex + convex" structure, while ours has a "convex + concave" structure. A natural extension of our work would be a full-fledged study of DC programming in the Wasserstein space $\tilde{\mathcal{F}} = \tilde{\mathcal{G}} - \tilde{\mathcal{H}}$ where $\tilde{\mathcal{G}}$ and $\tilde{\mathcal{H}}$ are convex along generalized geodesics and $\tilde{\mathcal{H}}$ is not necessarily potential energy $\mathcal{E}_H$. This problem possesses another implementation challenge regarding the Wasserstein gradient of $\tilde{\mathcal{H}}$.

Other works that bear a tangential relation to ours involve the use of forward-only Euler and ULA. Arbel et al. [7] study Wasserstein gradient flows with forward Euler for maximum mean discrepancy. This objective function is also nonconvex (specifically, weakly convex). Durmus et al. [26] analyze the ULA from the convex optimization perspective. Vempala et al. [67] show that LSI and Hessian boundedness suffice for fast convergence of the ULA where "fast" is understood as fast to the biased target since ULA is a biased algorithm. Balasubramanian et al. [9] analyze the ULA under quite mild conditions: log-density is Lipschitz/Hölder smooth. Bernton [11] studies the proximal-ULA also under the convex assumption, where the difference to the ULA is the first step: gradient descent is replaced by the proximal operator. Similar to ULA, proximal-ULA is asymptotically biased. To address nonsmoothness, another line of research utilizes Moreau-Yosida envelopes to create smooth approximations of the ULA dynamics [29, 50]. This approach is also applicable to certain classes of non-log-concave distributions [50] and is more of a flavour of discretization error quantification.

## 8 Conclusion

We propose a new semi FB Euler scheme as a discretization of Wasserstein gradient flow and show that it has favourably theoretical guarantees that the commonly used FB Euler does not yet have if the objective function is not convex along generalized geodesics. Our theoretical analysis opens up interesting avenues for future work. Given the ubiquity of nonconvexity, we hope that the idea can be reused in various contexts, such as with different optimal transport cost functions, different base spaces in the Wasserstein space [42], or submanifolds of the Wasserstein space (e.g., Bures-Wasserstein [42, 24]).

---

[1] Our code is available at `https://github.com/MCS-hub/OW24`

## Acknowledgments and Disclosure of Funding

This work is supported by the Research Council of Finland's Flagship programme: Finnish Center for Artificial Intelligence (FCAI), and additionally by grants 345811, 348952, and 346376 (VILMA: Virtual Laboratory for Molecular Level Atmospheric Transformations). The authors wish to thank the Finnish Computing Competence Infrastructure (FCCI) for supporting this project with computational and data storage resources. We thank the anonymous reviewers for their insightful comments and suggestions. H.P.H. Luu specifically thanks Michel Ledoux, Luigi Ambrosio, Alain Durmus, and Shuailong Zhu for helpful information.

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

# A  Theory

**Lemma 2** (Transfer lemma). *[3, Sect. 1] Let $T : \mathbb{R}^m \to \mathbb{R}^n$ be a measurable map, and $\mu \in \mathcal{P}(\mathbb{R}^m)$, then $T_{\#}\mu \in \mathcal{P}(\mathbb{R}^n)$ and $\int f(y)d(T_{\#}\mu)(y) = \int (f \circ T)(x)d\mu(x)$ for every measurable function $f : \mathbb{R}^n \to \mathbb{R}$, where the above identity has to be understood that: one of the integrals exits (potentially $\pm\infty$) iff the other one exists, and in such a case they are equal. Consequently, for a bounded function $f$, the above integrals exist as real numbers that are equal.*

**Lemma 3.** *[4, Rmk. 6.2.11] Let $\mu, \nu \in \mathcal{P}_{2,\mathrm{abs}}(X)$, then $T_{\nu}^{\mu} \circ T_{\mu}^{\nu} = I$ $\mu$-a.e. and $T_{\mu}^{\nu} \circ T_{\nu}^{\mu} = I$ $\nu$-a.e.*

**Theorem 6** (Characterization of Fréchet subdifferential for geodesically convex functions). *[4, Section 10.1] Suppose $\phi : \mathcal{P}_2(X) \to \mathbb{R} \cup \{+\infty\}$ is proper, l.s.c, convex on geodesics. Let $\mu \in \mathrm{dom}(\partial\phi) \cap \mathcal{P}_{2,\mathrm{abs}}(X)$, then a vector $\xi \in L^2(X, X, \mu)$ belongs to the Fréchet subdifferential of $\phi$ at $\mu$ if and only if*

$$\phi(\nu) - \phi(\mu) \geq \int_X \langle \xi(x), T_{\mu}^{\nu}(x) - x \rangle d\mu(x) \quad \forall \nu \in \mathrm{dom}(\phi).$$

**Lemma 4.** *Let $H : X \to \mathbb{R}$ be a convex function having quadratic growth and $\xi$ be a measurable selector of $\partial H$, i.e., $\xi(x) \in \partial H(x)$ for all $x \in X$. Then, for all $\mu \in \mathcal{P}_2(X)$,*

$$\int_X \|\xi(x)\|^2 d\mu(x) < +\infty. \tag{6}$$

*In other words, $\xi \in L^2(X, X, \mu)$ for all $\mu \in \mathcal{P}_2(X)$.*

*Proof.* Since $\xi(x) \in \partial H(x)$, by tangent inequality for convex functions, we have

$$H(y) \geq H(x) + \langle \xi(x), y - x \rangle, \quad \forall y \in X.$$

By picking $y = x + \epsilon\xi(x)$ for some $\epsilon > 0$, we get

$$H(x + \epsilon\xi(x)) - H(x) \geq \langle \xi(x), \epsilon\xi(x) \rangle = \epsilon\|\xi(x)\|^2. \tag{7}$$

Since $H$ has quadratic growth, for some $a > 0$,

$$\begin{aligned} H(x + \epsilon\xi(x)) - H(x) &\leq |H(x + \epsilon\xi(x))| + |H(x)| \\ &\leq a\left(\|x + \epsilon\xi(x)\|^2 + 1\right) + a(\|x\|^2 + 1) \\ &\leq 2a + a\|x\|^2 + a(\|x\| + \epsilon\|\xi(x)\|)^2 \\ &\leq 2a + a\|x\|^2 + 2a(\|x\|^2 + \epsilon^2\|\xi(x)\|^2) \\ &= 2a + 3a\|x\|^2 + 2a\epsilon^2\|\xi(x)\|^2. \end{aligned}$$

Combining with (7), it holds

$$\epsilon(1 - 2a\epsilon)\|\xi(x)\|^2 \leq 2a + 3a\|x\|^2.$$

By choosing $0 < \epsilon < 1/(2a)$, we obtain

$$\|\xi(x)\|^2 \leq \frac{2a}{\epsilon(1 - 2a\epsilon)} + \frac{3a}{\epsilon(1 - 2a\epsilon)}\|x\|^2.$$

Therefore, $\|\xi(x)\|^2$ has quadratic growth and - as a consequence - (6) holds for any $\mu \in \mathcal{P}_2(X)$. $\quad\square$

**Lemma 5.** *Let $\phi : \mathcal{P}_2(X) \to \mathbb{R} \cup \{+\infty\}$ be a proper function. Let $\mu^*$ be a local minimizer of $\phi$, then $\mu^*$ is a Fréchet stationary point of $\phi$.*

*Proof.* There exists $r > 0$ such that $\phi(\mu^*) \leq \phi(\mu)$ for all $\mu \in \mathcal{P}_2(X) : W_2(\mu, \mu^*) < r$. It follows that

$$\liminf_{\mu \xrightarrow{\text{Wass}} \mu^*} \frac{\phi(\mu) - \phi(\mu^*)}{W_2(\mu, \mu^*)} \geq 0,$$

so $0 \in \partial_F^- \phi(\mu^*)$, or $\mu^*$ is a Fréchet stationary point of $\phi$. $\quad\square$

**Lemma 6.** *Let $\phi : \mathcal{P}_2(X) \to \mathbb{R} \cup \{+\infty\}$ be a proper, l.s.c, geodesically convex function. Suppose that $\mu^* \in \mathcal{P}_{2,\mathrm{abs}}(X)$ is a Fréchet stationary point of $\phi$. Then, $\mu^*$ is a global minimizer of $\phi$.*

*Proof.* By definition of Fréchet stationarity, $0 \in \partial\phi(\mu^*)$. By characterization of subdifferential of geodesically convex functions (Thm. 6), it holds $\phi(\mu) \geq \phi(\mu^*)$ for all $\mu \in \mathrm{dom}(\phi)$, or $\mu^*$ is a global minimizer of $\phi$. $\quad\square$

**Lemma 7.** *Under Assumption 1, let $\mu^* \in \mathrm{dom}(\mathcal{F})$ be a local minimizer of $\mathcal{F}$, then $\mu^*$ is a critical point of $\mathcal{F}$, i.e., $\partial(\mathcal{H} + \mathcal{E}_G)(\mu^*) \cap \partial\mathcal{E}_H(\mu^*) \neq \emptyset$.*

*Proof.* Since $\mu^*$ is a local minimizer of $\mathcal{F}$, there exists $r > 0$ such that

$$\mathcal{F}(\mu^*) \leq \mathcal{F}(\mu), \quad \forall \mu \in \mathcal{P}_2(X) : W_2(\mu, \mu^*) < r. \tag{8}$$

Let $\xi$ be a measurable selector of $\partial H$. Thanks to Lemma 4, $\xi \in L^2(X, X, \mu^*)$. According to [5, Prop. 4.13], $\xi \in \partial\mathcal{E}_H(\mu^*)$. It follows from Thm. 6 that

$$\mathcal{E}_H(\mu) \geq \mathcal{E}_H(\mu^*) + \int_X \langle \xi(x), T^\mu_{\mu^*}(x) - x \rangle d\mu^*(x), \quad \forall \mu \in \mathcal{P}_2(X). \tag{9}$$

From (8) and (9), for $\mu \in B(\mu^*, r)$,

$$\mathcal{H}(\mu) + \mathcal{E}_G(\mu) \geq \mathcal{H}(\mu^*) + \mathcal{E}_G(\mu^*) + \int_X \langle \xi(x), T^\mu_{\mu^*}(x) - x \rangle d\mu^*(x).$$

Therefore, $\xi \in \partial(\mathcal{H} + \mathcal{E}_G)(\mu^*)$ since $\mathcal{H} + \mathcal{E}_G$ is geodesically convex. It follows that $\mu^*$ is a critical point of $\mathcal{F}$. $\quad\square$

**Lemma 8.** *Let $\mathcal{U}, \mathcal{V} : \mathcal{P}_2(X) \to \mathbb{R} \cup \{+\infty\}$. Assume that $\mathcal{U}$ and $\mathcal{V}$ are Fréchet subdifferentiable at $\mu$, the following statements hold*

    *a. $\partial^-_F(\mathcal{U} + \mathcal{V})(\mu) \supset \partial^-_F\mathcal{U}(\mu) + \partial^-_F\mathcal{V}(\mu)$.*

    *b. If $\mathcal{V}$ is Wasserstein differentiable of $\mu$, then*

$$\partial^-_F(\mathcal{U} + \mathcal{V})(\mu) = \partial^-_F\mathcal{U}(\mu) + \nabla_W\mathcal{V}(\mu). \tag{10}$$

*Proof.* Item a. is trivial from the definition of Fréchet subdifferential. For item b., from item a., we first see that,

$$\partial^-_F(\mathcal{U} + \mathcal{V})(\mu) \supset \partial^-_F\mathcal{U}(\mu) + \nabla_W\mathcal{V}(\mu). \tag{11}$$

On the other hand, we apply item a. for $\mathcal{U} + \mathcal{V}$ and $-\mathcal{V}$ to obtain

$$\partial^-_F\mathcal{U}(\mu) \supset \partial^-_F(\mathcal{U} + \mathcal{V})(\mu) + \partial^-_F(-\mathcal{V})(\mu).$$

Since $-\nabla_W\mathcal{V}(\mu) \in \partial^-_F(-\mathcal{V})(\mu)$, it follows that

$$\partial^-_F\mathcal{U}(\mu) \supset \partial^-_F(\mathcal{U} + \mathcal{V})(\mu) - \nabla_W\mathcal{V}(\mu). \tag{12}$$

From (11) and (12), we derive (10). $\quad\square$

**Lemma 9.** *Let $\mu \ll \mathscr{L}^d$, and $g$ is a strongly convex function. Then $\nabla g_{\#}\mu \ll \mathscr{L}^d$.*

*Proof.* Let $\Omega = \{x : \nabla^2 g(x) \text{ exists}\}$, then $\mathscr{L}^d(\mathbb{R}^d \setminus \Omega) = 0$ (Aleksandrov, see, e.g., [4, Thm. 5.5.4]). Since $g$ is strongly convex, $\nabla g$ is injective on $\Omega$ and $|\det \nabla^2 g| > 0$ on $\Omega$. By applying Lemma 5.5.3 [4], $\nabla g_{\#}\mu \ll \mathscr{L}^d$. $\quad\square$

**Lemma 10.** *[65] Let $\mathcal{G} : \mathcal{P}_2(X) \to \mathbb{R} \cup \{+\infty\}$ be proper and l.s.c. Suppose that $\mathcal{G}$ is convex along generalized geodesics. Let $\nu \in \mathcal{P}_{2,\mathrm{abs}}(X)$, $\mu, \pi \in \mathcal{P}_2(X)$. If $\xi \in \partial\mathcal{G}(\mu)$, then*

$$\int_X \langle \xi \circ T^\mu_\nu(x), T^\pi_\nu(x) - T^\mu_\nu(x) \rangle d\nu(x) \leq \mathcal{G}(\pi) - \mathcal{G}(\mu).$$

## A.1 Existence of a Borel measurable selector of the subdifferential of a convex function

Given a convex function $H : X \to \mathbb{R}$, we prove that there exists a Borel measurable selector $S(x) \in \partial H(x)$. Although this problem is of natural interest, we are not aware of it as well as its proof at least in standard textbooks in convex analysis. Credits go to a quite recent MathOverflow thread [35], from which we give detailed proof as follows.

Firstly, we recall Alexandroff's compactification of a topological space $(X, \tau)$. From set theory, $X$ is strictly smaller than $2^X$, which is the set of all subsets of $X$, i.e., there is no bijection from $X$ to $2^X$. So $2^X$ cannot be contained in $X$. So, there is an element named $\infty$ that is not in $X$. We denote $X^\infty = X \cup \{\infty\}$. One-point Alexandroff compactification states that (1) there exists a topology $\tau^\infty$ in $X^\infty$ accepting $X$ as a *topological subspace*, i.e., the original topology $\tau$ in $X$ is inherited from $\tau^\infty$, and (2) $(X^\infty, \tau^\infty)$ is compact.

The topology $\tau^\infty$ can be specifically described as follows: open sets of $\tau^\infty$ are either open sets of $\tau$ or the complements of the form $(X \setminus S) \cup \{\infty\}$ where $S$ are closed compact subsets of $X$.

In our case, $X = \mathbb{R}^d$ and $\tau$ is the standard Euclidean topology, the Alexandroff compactification of $X$ is also metrizable [17, Thm. 12.12]. It is, in fact, homeomorphic to the sphere $\mathbb{S}^d$ whose topology is inherited from the ambient space $\mathbb{R}^{d+1}$. Moreover, the mentioned metric is the Riemannian metric of $\mathbb{S}^d$ [49, Thm. 13.29].

Secondly, since $\psi(x) := H(x) + (1/2)\|x\|^2$ is 1-strongly convex, its Fenchel conjugate $y \mapsto \psi^*(y)$ defined as

$$\psi^*(y) = \sup_{x \in \mathbb{R}^d} \{\langle x, y \rangle - \psi(x)\}$$

is 1-smooth [72, Thm. 1].

By [64, Cor. 23.5.1], $(\partial \psi)^{-1} = \nabla \psi^*$ in the sense that

$$y \in \partial \psi(\nabla \psi^*(y)) \quad \forall y \in \mathbb{R}^d. \tag{13}$$

On the other hand, $\partial \psi$ is *strongly monotone* [54, Ex. 3.9] in the following sense,

$$\langle x_2 - x_1, y_2 - y_1 \rangle \geq \|x_2 - x_1\|^2 \tag{14}$$

for all $x_1, x_2 \in \mathbb{R}^d$ and $y_1 \in \partial \psi(x_1), y_2 \in \partial \psi(x_2)$.

We see that $\nabla \psi^*$ is subjective. Indeed we show that for every $x \in \mathbb{R}^d$, there exists $y \in \mathbb{R}^d$ such that $\nabla \psi^*(y) = x$. We show that relation holds for any $y \in \partial \psi(x)$. By contradiction, suppose that $\nabla \psi^*(y) \neq x$. From the strong monotonicity of $\partial \psi$ as in (14), $\partial \psi(\nabla \psi^*(y)) \cap \partial \psi(x) = \emptyset$. However, from (13) and by the choice of $y$, it holds $y \in \partial \psi(\nabla \psi^*(y)) \cap \partial \psi(x)$. This is a contradiction.

Thirdly, we recall a fundamental result on the compactness of the subdifferential of a convex function: if $C$ is compact, then $\partial \psi(C)$ is compact [64, Thm. 24.7].

Fourthly, we need the Federer-Morse theorem [13] as follows:

**Theorem 7.** *Let $Z$ be a compact metric space, $Y$ be a Hausdorff topological space and $f : Z \to Y$ be a continuous mapping. Then, there exists a Borel set $B \subset Z$ such that $f(B) = f(Z)$ and $f$ is injective on $B$. Furtheremore, $f^{-1} : f(Z) \to B$ is Borel.*

Now we observe that $\nabla \psi^*(x) \to \infty$ was $x \to \infty$. Otherwise, by using the compactness of $\partial \psi$ and (13), we will get a contradiction immediately. We then can extend $\nabla \psi^*$ in a *continuous* way in the Alexandorff compactification space $X^\infty = \mathbb{R}^d \cup \{\infty\}$ by simply putting $\nabla \psi^*(\infty) = \infty$. We shall show that this extension of $\nabla \psi^*$ is continuous from $(X^\infty, \tau^\infty)$ to $(X^\infty, \tau^\infty)$, or $(\nabla \psi^*)^{-1}(V) \in \tau^\infty$ for all $V \in \tau^\infty$. Recall that, by construction, open sets of $\tau^\infty$ are either open sets of $\tau$ or the complements of the form $(X \setminus S) \cup \{\infty\}$ where $S$ are compact subsets of $X$. The former type of open sets is handled easily since $\nabla \psi^*$ is already continuous in $(X, \tau)$. For the latter type, let $U = (\nabla \psi^*)^{-1}((X \setminus S) \cup \{\infty\}) = (\nabla \psi^*)^{-1}(X \setminus S) \cup \{\infty\} = (X \setminus (\nabla \psi^*)^{-1}(S)) \cup \{\infty\}$ for a compact set $S$. Proving $U$ open boils down to proving $(\nabla \psi^*)^{-1}(S)$ compact. Indeed, it is closed since $S$ is closed. It is bounded. Otherwise, it will be contradictory to $\nabla \psi^*(x) \to \infty$ as $x \to \infty$.

We now can apply the Federer-Morse theorem for $Z = Y = (X^\infty, \tau^\infty)$ by noting that $(X^\infty, \tau^\infty)$ is metrizable and a metric space is a Hausdorff space, and for $f = \nabla \psi^*$: there exists a Borel set $B \subset$

$X^\infty$ such that $\nabla \psi^*|_B : B \to X^\infty$ is a bijection and the inverse mapping $(\nabla \psi^*|_B)^{-1} : X^\infty \to B$ is Borel measurable (here Borel set/measurability are with respect to $\tau^\infty$, not yet $\tau$). This is the Borel (w.r.t. $\tau^\infty$) selector of $\partial \psi$.

Finally, we need to convert Borel measurability w.r.t. $\tau^\infty$ to Borel measurability w.r.t. $\tau$. In terms of mapping, $\infty$ is mapped to $\infty$ either way around. So we only need to show: $(\nabla \psi^*|_B)^{-1} : X \to X$ is Borel measurable w.r.t. $\tau$. Take any Borel set (w.r.t. $\tau$) $E \subset X$, $(\nabla \psi^*|_B)(B \cap E)$ is Borel set w.r.t. $\tau^\infty$ and does not contain $\infty$. We shall prove $(\nabla \psi^*|_B)(B \cap E)$ is a Borel set w.r.t. $\tau$. This follows directly from the following claim, which is from another Mathematics Stack Exchange thread [19].

Claim [19]:

$$\sigma(\tau^\infty) = \sigma(\tau \cup \{\infty\}) = \sigma(\tau) \cup \{V \cup \{\infty\} : V \in \sigma(\tau)\}. \tag{15}$$

A sketch of the claim proof goes as follows. For the first equality in (15), first we have $\sigma(\tau \cup \{\infty\}) \subset \sigma(\tau^\infty)$ because (1) $\tau \subset \tau^\infty$ and (2) $\{\infty\} = X^\infty \setminus X \in \sigma(\tau^\infty)$ as $X, X^\infty \in \tau^\infty$. On the other hand, $\sigma(\tau^\infty) \subset \sigma(\tau \cup \{\infty\})$ because, again, of the construction of $\tau^\infty$: let $U \in \tau^\infty$, if $U \in \tau$ then $U \in \sigma(\tau \cup \{\infty\})$, otherwise $U = (X \setminus S) \cup \{\infty\}$ for some compact set $S \subset X$. As $X = \mathbb{R}^d$, $S$ is closed, so $(X \setminus S) \in \tau$ implying $U \in \sigma(\tau \cup \{\infty\})$. For the second equality in (15), it is straight forward to verify that $\mathscr{G} := \sigma(\tau) \cup \{V \cup \{\infty\} : V \in \sigma(\tau)\}$ is a sigma-algebra in $X^\infty$. Since $\tau \cup \{\infty\} \subset \mathscr{G}$, it holds $\sigma(\tau \cup \{\infty\}) \subset \mathscr{G}$. Conversely, as $\sigma(\tau) \subset \sigma(\tau \cup \{\infty\})$ and - consequently - $V \cup \{\infty\} \in \sigma(\tau \cup \{\infty\})$ for all $V \in \sigma(\tau)$, it holds $\mathscr{G} \subset \sigma(\tau \cup \{\infty\})$.

We conclude that $(\nabla \psi^*|_B)^{-1} : X \to X$ is a Borel (w.r.t. $\tau$) selector of $\partial \psi$. As a consequence, $S := (\nabla \psi^*|_B)^{-1} - I$ is a Borel measurable selector of $\partial H$.

## A.2  Maximum Mean Discrepancy

### A.2.1  Definition

The notion of Maximum Mean Discrepancy (MMD) between two distributions is introduced in [34]. Given a kernel $k : X \times X \to \mathbb{R}$ and denote by $\mathcal{K}$ its reproducing kernel Hilbert space. Given two distributions $\mu$ and $\nu$, the maximum mean discrepancy between them are defined as

$$\mathrm{D}_{\mathrm{MML}}(\mu, \nu) = \|f_{\mu,\nu}\|_{\mathcal{K}}, \quad \text{where } f_{\mu,\nu}(z) = \int k(x, z) d\nu(x) - \int k(x, z) d\mu(x), \ \forall z \in X,$$

where $f_{\mu,\nu}$ is called the witness function. Given some target (or optimal) distribution $\mu^*$, we seek to optimize $\mathcal{F}(\mu) = \frac{1}{2} \mathrm{D}_{\mathrm{MML}}(\mu, \mu^*)^2$. Note that $\mathcal{F}(\mu)$ admits the following free-energy expression [7]

$$\mathcal{F}(\mu) = \int F(x) d\mu(x) + \frac{1}{2} \int k(x, x') d\mu(x) d\mu(x') + C$$

where

$$F(x) = - \int k(x, x') d\mu^*(x'), \quad C = \frac{1}{2} \int k(x, x') d\mu^*(x) d\mu^*(x').$$

In general, $\mathcal{F}$ is not convex along generalized geodesics. Rather, it exhibits some weakly convex structure [7] that falls within the DC spectrum as detailed in Subsection A.2.3.

### A.2.2  Connection with infinite-width one hidden layer neural networks

See also [7, 65]. We include the discussion here for completeness.

Consider a one-hidden-layer neural network

$$f(x) = \frac{1}{n} \sum_{i=1}^{n} \psi(x, z_i)$$

where $\psi(x, z_i) = w_i \sigma(\langle x, \theta_i \rangle)$, $z_i = (\theta_i, w_i) \in Z$ is the parameters (in and out) associated with $i$-th hidden neuron, $n$ is the number of hidden neurons, $\sigma$ is the activation function.

When the number of hidden neurons $n$ tends to infinity, $f$ can be written as $f(x) = \int \psi(x, z) d\mu(z)$ for some distribution $\mu$ over the parameter space $Z$. Given data $(x, y) \sim p(x, y)$ and assuming quadratic loss, the optimization problem reads

$$\min_{\mu \in \mathcal{P}_2(Z)} \mathcal{L}(\mu) := \mathbb{E}_{(x,y) \sim p(x,y)} \left( y - \int \psi(x, z) d\mu(z) \right)^2.$$

We assume that the model is well-specified, i.e., there exists $\mu^* \in \mathcal{P}_2(Z)$ such that

$$\mathbb{E}_{y \sim p(y|x)}(y) = \int \psi(x, z) d\mu^*(z). \tag{16}$$

Expanding $\mathcal{L}$, we get

$$\mathcal{L}(\mu) = -2 \int \mathbb{E}_{(x,y) \sim p(x,y)}(y\psi(x, z)) d\mu(z) + \iint \mathbb{E}_{x \sim p(x)}(\psi(x, z)\psi(x, z')) d\mu(z) d\mu(z') + C$$

where $C$ does not depend on $\mu$.

Naturally, we define the kernel

$$k(z, z') = \mathbb{E}_{x \sim p(x)}(\psi(x, z)\psi(x, z')) \quad \forall z, z' \in Z.$$

Using the assumption that the model is well-specified (16), we obtain

$$\begin{aligned}
\int k(z, z') d\mu^*(z') &= \int \mathbb{E}_{x \sim p(x)}(\psi(x, z)\psi(x, z')) d\mu^*(z') \\
&= \mathbb{E}_{x \sim p(x)} \left( \int \psi(x, z)\psi(x, z') d\mu^*(z') \right) \\
&= \mathbb{E}_{x \sim p(x)} \left( \psi(x, z) \int \psi(x, z') d\mu^*(z') \right) \\
&= \mathbb{E}_{x \sim p(x)} \left( \psi(x, z) \mathbb{E}_{y \sim p(y|x)}(y) \right) \\
&= \mathbb{E}_{(x,y) \sim p(x,y)} \left( y\psi(x, z) \right).
\end{aligned}$$

Therefore, $\mathcal{L}$ can be written as

$$\mathcal{L}(\mu) = -2 \int \left[ \int k(z, z') d\mu^*(z') \right] d\mu(z) + \iint k(z, z') d\mu(z) d\mu(z') + C.$$

This is exactly the MMD setting in Sect. A.2.1.

### A.2.3  DC structure

Let $k$ be a kernel whose gradient is Lipschitz continuous, i.e., for some $L > 0$,

$$\|\nabla k(x, y) - \nabla k(x', y')\| \leq L \left( \|x - x'\|^2 + \|y - y'\|^2 \right)^{\frac{1}{2}}, \quad \forall x, x', y, y' \in X,$$

which can be expressed equivalently as

$$\|\nabla_x k(x, y) - \nabla_x k(x', y')\|^2 + \|\nabla_y k(x, y) - \nabla_y k(x', y')\|^2 \leq L^2 \left( \|x - x'\|^2 + \|y - y'\|^2 \right).$$

Let $\mu^*$ be some target distribution and consider the free-energy functional

$$\mathcal{F}(\mu) = \iint k(x, y) d\mu(x) d\mu(y) - 2 \iint k(x, y) d\mu^*(y) d\mu(x).$$

Let $\alpha > 0$, we can rewrite $\mathcal{F}$ as follows

$$\mathcal{F}(\mu) = \iint \left[ \alpha\|x\|^2 + \alpha\|y\|^2 + k(x, y) \right] d\mu(x) d\mu(y) - 2 \int \left[ \alpha\|x\|^2 + \int k(x, y) d\mu^*(y) \right] d\mu(x).$$

As $k$ is Lipschitz smooth w.r.t. $(x, y)$, $x \mapsto \int k(x, y)d\mu^*(y)$ is also Lipschitz smooth. Indeed,

$$\left\| \nabla_x \int k(x, y)d\mu^*(y) - \nabla_x \int k(x', y)d\mu^*(y) \right\|$$

$$= \left\| \int (\nabla_x k(x, y) - \nabla_x k(x', y))d\mu^*(y) \right\|$$

$$\leq \left( \int \|\nabla_x k(x, y) - \nabla_x k(x', y)\|^2 \, d\mu^*(y) \right)^{\frac{1}{2}}$$

$$\leq L \left( \int \|x - x'\|^2 \, d\mu^*(y) \right)^{\frac{1}{2}}$$

$$= L\|x - x'\|.$$

Next, as a standard result, if $f$ is an L-smooth function, $x \mapsto (\alpha/2)\|x\|^2 \pm f(x)$ are convex whenever $\alpha \geq L$. Therefore, for $\alpha \geq L$, $W(x, y) := \alpha\|x\|^2 + \alpha\|y\|^2 + k(x, y)$ is convex and $F(x) = -2\left[\alpha\|x\|^2 + \int_X k(x, y)d\mu^*(y)\right]$ is concave. From [4, Prop. 9.3.5], the interaction energy corresponding to $W$ is generalized geodesically convex.

## A.3 Proof of Lemma 1

Since $I + \eta S$ is a subgradient selector of a convex function, the optimal transport between $\mu_n$ and $\nu_{n+1}$ is given by

$$T_{\mu_n}^{\nu_{n+1}} = I + \eta S. \tag{17}$$

and between $\mu_{n+1}$ and $\nu_{n+1}$ [4, Lem. 10.1.2]

$$T_{\mu_{n+1}}^{\nu_{n+1}} \in I + \eta \partial \left(\mathcal{E}_G + \mathcal{H}\right)(\mu_{n+1}). \tag{18}$$

Since $\mathcal{E}_H$ is convex along generalized geodesics [4, Prop. 9.3.2] and $S$ is a subgradient of $\mathcal{E}_H$ at $\mu_n$ [5, Proposition 4.13], by Lem. 10 it holds, for any $\nu \in \mathcal{P}_{2,\mathrm{abs}}(X)$,

$$\mathcal{E}_H(\mu_{n+1}) \geq \mathcal{E}_H(\mu_n) + \int_X \langle S \circ T_\nu^{\mu_n}(x), T_\nu^{\mu_{n+1}}(x) - T_\nu^{\mu_n}(x)\rangle d\nu(x).$$

By choosing $\nu = \nu_{n+1}$ (note that $\nu_{n+1} \in \mathcal{P}_{2,\mathrm{abs}}(X)$),

$$\mathcal{E}_H(\mu_{n+1}) \geq \mathcal{E}_H(\mu_n) + \int_X \langle S \circ T_{\nu_{n+1}}^{\mu_n}(x), T_{\nu_{n+1}}^{\mu_{n+1}}(x) - T_{\nu_{n+1}}^{\mu_n}(x)\rangle d\nu_{n+1}(x)$$

$$= \mathcal{E}_H(\mu_n) + \frac{1}{\eta} \int_X \langle (T_{\mu_n}^{\nu_{n+1}} - I) \circ T_{\nu_{n+1}}^{\mu_n}(x), T_{\nu_{n+1}}^{\mu_{n+1}}(x) - T_{\nu_{n+1}}^{\mu_n}(x)\rangle d\nu_{n+1}(x)$$

$$= \mathcal{E}_H(\mu_n) + \frac{1}{\eta} \int_X \langle x - T_{\nu_{n+1}}^{\mu_n}(x), T_{\nu_{n+1}}^{\mu_{n+1}}(x) - T_{\nu_{n+1}}^{\mu_n}(x)\rangle d\nu_{n+1}(x) \tag{19}$$

where the second equality uses (17) and the last one uses Lem. 3.

On the other hand, since $\mathcal{E}_G + \mathcal{H}$ is convex along generalized geodesics, by applying Lem. 10 for $\mathcal{E}_G + \mathcal{H}$ at $\mu_{n+1}$ with a subgradient $\eta^{-1}(T_{\mu_{n+1}}^{\nu_{n+1}} - I) \in \partial(\mathcal{E}_G + \mathcal{H})(\mu_{n+1})$ (from (18)),

$$(\mathcal{E}_G + \mathcal{H})(\mu_n)$$

$$\geq (\mathcal{E}_G + \mathcal{H})(\mu_{n+1}) + \int_X \left\langle \frac{(T_{\mu_{n+1}}^{\nu_{n+1}} - I)}{\eta} \circ T_{\nu_{n+1}}^{\mu_{n+1}}(x), T_{\nu_{n+1}}^{\mu_n}(x) - T_{\nu_{n+1}}^{\mu_{n+1}}(x) \right\rangle d\nu_{n+1}(x)$$

$$= (\mathcal{E}_G + \mathcal{H})(\mu_{n+1}) + \frac{1}{\eta} \int_X \left\langle x - T_{\nu_{n+1}}^{\mu_{n+1}}(x), T_{\nu_{n+1}}^{\mu_n}(x) - T_{\nu_{n+1}}^{\mu_{n+1}}(x) \right\rangle d\nu_{n+1}(x) \tag{20}$$

where the last equality uses Lem. 3.

By adding (20) and (19) side by side,

$$\mathcal{F}(\mu_n) \geq \mathcal{F}(\mu_{n+1}) + \frac{1}{\eta} \int_X \|T_{\nu_{n+1}}^{\mu_n}(x) - T_{\nu_{n+1}}^{\mu_{n+1}}(x)\|^2 d\nu_{n+1}(x). \tag{21}$$

## A.4  Proof of Theorem 1

Let $\mu^* \in \mathcal{P}_2(X)$ be a cluster point of $\{\mu_n\}_{n\in\mathbb{N}}$. There exists a subsequence $\mu_{n_k} \xrightarrow{\text{Wass}} \mu^*$. It holds

$$
\begin{aligned}
\liminf_{k\to\infty} \mathcal{F}(\mu_{n_k}) &= \liminf_{k\to\infty} \left( \mathscr{H}(\mu_{n_k}) + \mathcal{E}_F(\mu_{n_k}) \right) \\
&= \liminf_{k\to\infty} \mathscr{H}(\mu_{n_k}) + \mathcal{E}_F(\mu^*) \\
&\geq \mathscr{H}(\mu^*) + \mathcal{E}_F(\mu^*),
\end{aligned}
$$

since $\mathscr{H}$ is l.s.c. and $\mathcal{E}_F$ is continuous w.r.t. Wasserstein topology. Therefore, $\mathscr{H}(\mu^*) < +\infty$, which further implies that $\mu^* \in \mathcal{P}_{2,\text{abs}}(X)$.

We have

$$
\begin{aligned}
\int_X \| T^{\mu_n}_{\nu_{n+1}}(x) - T^{\mu_{n+1}}_{\nu_{n+1}}(x) \|^2 d\nu_{+1}(x) &= \int_X \| T^{\mu_n}_{\nu_{n+1}}(x) - T^{\mu_{n+1}}_{\nu_{n+1}}(x) \|^2 dT^{\nu_{n+1}}_{\mu_n}{}_\# \mu_n(x) \\
&= \int_X \| x - T^{\mu_{n+1}}_{\nu_{n+1}} \circ T^{\nu_{n+1}}_{\mu_n}(x) \|^2 d\mu_n(x). \quad (22)
\end{aligned}
$$

We observe that $T^{\mu_{n+1}}_{\nu_{n+1}} \circ T^{\nu_{n+1}}_{\mu_n}$ is a (possibly non-optimal) transport pushing $\mu_n$ to $\mu_{n+1}$, by the optimality of $T^{\mu_{n+1}}_{\mu_n}$,

$$
\int_X \| x - T^{\mu_{n+1}}_{\nu_{n+1}} \circ T^{\nu_{n+1}}_{\mu_n}(x) \|^2 d\mu_n(x) \geq \int_X \| x - T^{\mu_{n+1}}_{\mu_n}(x) \|^2 d\mu_n(x) = W_2^2(\mu_n, \mu_{n+1}). \quad (23)
$$

By Lem. 1 and (22), (23),

$$
\mathcal{F}(\mu_n) \geq \mathcal{F}(\mu_{n+1}) + \frac{1}{\eta} W_2^2(\mu_n, \mu_{n+1}). \quad (24)
$$

Note that $\mathcal{F}$ is bounded below (Assumption 1), telescoping (24) gives us

$$
\sum_{n=0}^{\infty} W_2^2(\mu_n, \mu_{n+1}) < +\infty. \quad (25)
$$

In particular, $W_2(\mu_n, \mu_{n+1}) \to 0$. This together with $\mu_{n_k} \xrightarrow{\text{Wass}} \mu^*$ implies $\mu_{n_k+1} \xrightarrow{\text{Wass}} \mu^*$.

Under Assumption 3, $S = \nabla H$ and $S$ is continuous. Next, recall $\nu_{n_k+1} = (I + \eta S)_\# \mu_{n_k}$, we show

$$
\nu_{n_k+1} \xrightarrow{\text{narrow}} \nu^* := (I + \eta S)_\# \mu^* \text{ as } k \to +\infty. \quad (26)
$$

Thus, let $f$ be a continuous and bounded test functional in $X$, by using transfer lemma 2,

$$
\begin{aligned}
\lim_{k\to\infty} \int_X f(x) d\nu_{n_k+1}(x) &= \lim_{k\to\infty} \int_X f(x) d(I + \eta S)_\# \mu_{n_k}(x) \\
&= \lim_{k\to\infty} \int_X f(x + \eta S(x)) d\mu_{n_k}(x) \quad (27) \\
&= \int_X f(x + \eta S(x)) d\mu^*(x) \\
&= \int_X f(x) d\nu^*(x), \quad (28)
\end{aligned}
$$

since $S$ is continuous. So $\nu_{n_k+1} \xrightarrow{\text{narrow}} \nu^*$. We go one step further and prove that $\nu_{n_k+1}$ actually converges to $\nu^*$ in the Wasserstein metric. This boils down to showing convergence in second-order moments, i.e.,

$$
\mathfrak{m}_2(\nu_{n_k+1}) \to \mathfrak{m}_2(\nu^*),
$$

which is equivalent to showing that

$$
\int_X \| x + \eta S(x) \|^2 d\mu_{n_k}(x) \to \int_X \| x + \eta S(x) \|^2 d\mu^*(x).
$$

On the other hand, $\psi(x) := \|x + \eta S(x)\|^2$ has quadratic growth (follows from Lem. 4) and $\mu_{n_k} \to \mu^*$ in the Wasserstein metric, so [3, Prop. 2.4]

$$\lim_{k \to \infty} \int_X \|x + \eta S(x)\|^2 d\mu_{n_k}(x) = \int_X \|x + \eta S(x)\|^2 d\mu^*(x)$$

Therefore, $\nu_{n_k+1} \to \nu^*$ in Wasserstein metric.

To proceed further, we need the following theorem stating that the graph of the subdifferential of a geodesically convex function is closed under the product of Wasserstein and *weak* topologies.

**Theorem 8** (Closedness of subdifferential graph). *[4, Lemma 10.1.3] Let $\phi$ be a geodesically convex functional satisfying* $\mathrm{dom}(\partial \phi) \subset \mathcal{P}_{2,\mathrm{abs}}(X)$. *Let* $\{\mu_n\}_{n \in \mathbb{N}}$ *be a sequence converging in Wasserstein metric to* $\mu \in \mathrm{dom}(\phi)$. *Let* $\xi_n \in \partial \phi(\mu_n)$ *be satisfying*

$$\sup_{n \in \mathbb{N}} \int_X \|\xi_n(x)\|^2 d\mu_n(x) < +\infty, \tag{29}$$

*and converging* weakly *to* $\xi \in L^2(X, X, \mu)$ *in the following sense:*

$$\lim_{n \to \infty} \int_X \zeta(x)\xi_n(x)d\mu_n(x) = \int_X \zeta(x)\xi(x)d\mu(x), \quad \forall \zeta \in C_c^\infty(X). \tag{30}$$

*Then* $\xi \in \partial \phi(\mu)$.

As a side note, we need the notion of weak convergence in the above theorem because – unlike subdifferentials in flat Euclidean space – each $\xi_n$ lives in its own $L^2(X, X, \mu_n)$ space.

Back to our proof, for item (29), we show that

$$\sup_{n \in \mathbb{N}} \int_X \left\| T_{\mu_n}^{\nu_n}(x) - x \right\|^2 d\mu_n(x) < +\infty. \tag{31}$$

We proceed as follows to prove (31). We first show that $\sup_{n \in \mathbb{N}} \mathfrak{m}_2(\mu_n) < +\infty$. By contradiction, by assuming $\sup_{n \in \mathbb{N}} \mathfrak{m}_2(\mu_n) = +\infty$, we can extract a subsequence $\{\mu_{n_k}\}_{k \in \mathbb{N}}$ such that

$$\lim_{k \to \infty} \mathfrak{m}_2(\mu_{n_k}) = +\infty. \tag{32}$$

By compactness assumption, there further exists a subsequence $\{\mu_{n_{k_i}}\}_{i \in \mathbb{N}}$ such that $\mu_{n_{k_i}}$ converges (in Wasserstein metric) to some $\mu^{**} \in \mathcal{P}_2(X)$, implying that $\lim_{i \to \infty} \mathfrak{m}_2(\mu_{n_{k_i}}) = \mathfrak{m}_2(\mu^{**})$, which contradicts (32). Therefore, $\sup_{n \in \mathbb{N}} \mathfrak{m}_2(\mu_n) < +\infty$. We next show that $\sup_{n \in \mathbb{N}} \mathfrak{m}_2(\nu_n) < +\infty$. Indeed, as $\|S\|^2$ has quadratic growth,

$$\|S(x)\|^2 \le c(\|x\|^2 + 1)$$

for some $c > 0$. So

$$\begin{aligned} \mathfrak{m}_2(\nu_{n+1}) &= \int_X \|x\|^2 d\nu_{n+1}(x) \\ &= \int_X \|x\|^2 d(I + \eta S)_\# \mu_n(x) \\ &= \int_X \|x + \eta S(x)\|^2 d\mu_n(x) \\ &\le 2 \int_X \|x\|^2 d\mu_n(x) + 2\eta^2 \int_X \|S(x)\|^2 d\mu_n(x) \\ &\le (2 + 2c\eta^2)\mathfrak{m}_2(\mu_n) + 2\eta^2 c, \end{aligned}$$

implying that $\sup_{n \in \mathbb{N}} \mathfrak{m}_2(\nu_n) < +\infty$. This in conjunction with $G$ having quadratic growth implies that $\sup_{n \in \mathbb{N}} |\mathcal{E}_G(\nu_n)| < +\infty$. Furthermore, $\inf_n (\mathcal{E}_G + \mathscr{H})(\mu_n) > -\infty$ otherwise by lower semicontinuity of $\mathscr{H}$ and compactness of $\{\mu_n\}_{n \in \mathbb{N}}$ we get a contradiction.

Now, as $\eta^{-1}(T_{\mu_{n+1}}^{\nu_{n+1}} - I) \in \partial (\mathcal{E}_G + \mathscr{H})(\mu_{n+1})$ and $\mathcal{E}_G + \mathscr{H}$ is geodesically convex, by applying Thm. 6, it holds

$$(\mathcal{E}_G + \mathscr{H})(\nu_{n+1}) \ge (\mathcal{E}_G + \mathscr{H})(\mu_{n+1}) + \frac{1}{\eta} \int_X \|T_{\mu_{n+1}}^{\nu_{n+1}}(x) - x\|^2 d\mu_{n+1}(x). \tag{33}$$

The finiteness as in (31) then follows from $\sup_{n \in \mathbb{N}} |\mathcal{E}_G(\nu_n)| < +\infty, \sup_{n \in \mathbb{N}} -(\mathcal{E}_G + \mathscr{H})(\mu_n) < +\infty$ as proved and $\sup_{n \in \mathbb{N}} \mathscr{H}(\nu_n) < +\infty$ as assumed.

We next prove that there is a subsequence of $\{\mu_{n_k}\}_{k \in \mathbb{N}}$ such that

$$T_{\mu_{n_{k_j}+1}}^{\nu_{n_{k_j}}+1} - I \to T_{\mu^*}^{\nu^*} - I \text{ weakly.} \tag{34}$$

We consider the sequence of optimal plans between $\mu_n$ and $\nu_n$ as follows

$$\rho_n = (I, T_{\mu_n}^{\nu_n})_\# \mu_n, \quad \forall n \in \mathbb{N}.$$

We observe that

$$\mathfrak{m}_2(\rho_n) = \int_{X \times X} (\|x\|^2 + \|y\|^2) d\rho_n(x,y) = \int_X \|x\|^2 d\mu_n(x) + \int_X \|y\|^2 d\nu_n(y) < +\infty,$$

so $\rho_n \in \mathcal{P}_2(X \times X)$ for all $n \in \mathbb{N}$. Since $\sup_{n \in \mathbb{N}} \mathfrak{m}_2(\nu_n) < +\infty$ as proved, and as a Wasserstein ball is relatively compact under narrow topology, $\{\nu_n\}_{n \in \mathbb{N}}$ is relatively compact under narrow topology. The same property holds for $\{\mu_n\}_{n \in \mathbb{N}}$. According to Prokhorov [3, Theorem 1.3], $\{\mu_n\}_{n \in \mathbb{N}}$ and $\{\nu_n\}_{n \in \mathbb{N}}$ are tight. By [3, Remark 1.4], $\{\rho_n\}_{n \in \mathbb{N}}$ is also tight, hence relatively compact under narrow topology in $\mathcal{P}_2(X \times X)$. Consequently, $\{\rho_{n_k+1}\}_{k \in \mathbb{N}}$ admits a subsequence converging narrowly to some $\rho^* \in \mathcal{P}(X \times X)$. Let's say

$$\rho_{n_{k_i}+1} \xrightarrow{\text{narrow}} \rho^* \text{ as } i \to \infty. \tag{35}$$

We can see that $\text{proj}_{1\#} \rho^* = \mu^*, \text{proj}_{2\#} \rho^* = \nu^*$. We further show that $\rho^* \in \mathcal{P}_2(X \times X)$, or equivalently,

$$\int_{X \times X} (\|x\|^2 + \|y\|^2) d\rho^*(x,y) < +\infty. \tag{36}$$

Let $C \in \mathbb{N}$, thanks to the narrow convergence in (35), we have

$$\int_{X \times X} \min\{\|x\|^2 + \|y\|^2, C\} d\rho_{n_{k_i}+1}(x,y) \to \int_{X \times X} \min\{\|x\|^2 + \|y\|^2, C\} d\rho^*(x,y).$$

Furthermore,

$$\int_{X \times X} \min\{\|x\|^2 + \|y\|^2, C\} d\rho_{n_{k_i}+1}(x,y) \leq \sup_{n \in \mathbb{N}} \mathfrak{m}_2(\mu_n) + \sup_{n \in \mathbb{N}} \mathfrak{m}_2(\nu_n) := M < +\infty.$$

Passing to the limit, we get

$$\int_{X \times X} \min\{\|x\|^2 + \|y\|^2, C\} d\rho^*(x,y) \leq M$$

for all $C \in \mathbb{N}$. Sending $C$ to $\infty$ and applying Monotone Convergence Theorem we derive (36).

Back to the main proof, since $\{\rho_{n_{k_i}+1}\}_{i \in \mathbb{N}}$ is a sequence of optimal plans, its limit, $\rho^*$ is also optimal [3, Proposition 2.5]. Therefore,

$$\rho^* = (I, T_{\mu^*}^{\nu^*})_\# \mu^*.$$

Moreover, as

$$\mathfrak{m}_2(\rho_{n_{k_i}+1}) = \mathfrak{m}_2(\mu_{n_{k_i}+1}) + \mathfrak{m}_2(\nu_{n_{k_i}+1}) \to \mathfrak{m}_2(\mu^*) + \mathfrak{m}_2(\nu^*) = \mathfrak{m}_2(\rho^*),$$

we have

$$\rho_{n_{k_i}+1} \xrightarrow{\text{Wass}} \rho^* \text{ as } i \to \infty.$$

Now let's take any test function $\zeta \in C_c^\infty(X)$, we show

$$\lim_{j \to \infty} \int_X \zeta(x) T_{\mu_{n_{k_j}+1}}^{\nu_{n_{k_j}}+1}(x) d\mu_{n_{k_j}+1}(x) = \int_X \zeta(x) T_{\mu^*}^{\nu^*}(x) d\mu^*(x). \tag{37}$$

Indeed, $(x, y) \mapsto \zeta(x) \operatorname{proj}_i(y)$ where $\operatorname{proj}_i$ is the projection into the $i$-th coordinate is continuous and has quadratic growth since $\zeta(x)$ is bounded and $\operatorname{proj}_i(y)$ is linear.

Since $\rho_{n_{k_j}+1} \xrightarrow{\text{Wass}} \rho^*$, it holds: for each $i \in [d]$,

$$\lim_{j \to \infty} \int_X \zeta(x) \operatorname{proj}_i \left( T_{\mu_{n_{k_j}}+1}^{\nu_{n_{k_j}}+1}(x) \right) d\mu_{n_{k_j}}+1(x)$$

$$= \lim_{j \to \infty} \int_X \zeta(x) \operatorname{proj}_i(y) d(I, T_{\mu_{n_{k_j}}+1}^{\nu_{n_{k_j}}+1})_{\#}\mu_{n_{k_j}}+1(x, y)$$

$$= \lim_{j \to \infty} \int_X \zeta(x) \operatorname{proj}_i(y) d\rho_{n_{k_j}}+1(x, y)$$

$$= \int_X \zeta(x) \operatorname{proj}_i(y) d\rho^*(x, y)$$

$$= \int_X \zeta(x) \operatorname{proj}_i(y) d(I, T_{\mu^*}^{\nu^*})_{\#}\mu^*(x, y)$$

$$= \int_X \zeta(x) \operatorname{proj}_i(T_{\mu^*}^{\nu^*}(x)) d\mu^*(x),$$

so (37) holds. Consequently, (34) also holds by noticing that

$$\int_X x\zeta(x) d\mu_{n_{k_j}}+1(x) \to \int_X x\zeta(x) d\mu^*(x) \text{ as } j \to \infty.$$

By the closedness of subdifferential graph of $\partial(\mathcal{E}_G + \mathcal{H})$ (Thm. 8), we obtain

$$\frac{T_{\mu^*}^{\nu^*} - I}{\eta} \in \partial(\mathcal{E}_G + \mathcal{H})(\mu^*). \tag{38}$$

On the other hand, by the definition of $\nu^*$ in (26), we get $T_{\mu^*}^{\nu^*} = I + \eta S$. Together with (38), $S \in \partial(\mathcal{E}_G + \mathcal{H})(\mu^*)$. Noting that $S \in \partial\mathcal{E}_H(\mu^*)$, it holds $S \in \partial(\mathcal{E}_G + \mathcal{H})(\mu^*) \cap \partial\mathcal{E}_H(\mu^*)$ and we conclude $\mu^*$ is a critical point of $\mathcal{F}$.

### A.5  Proof of Theorem 2

We see that

$$\|\mathcal{G}_\eta(\mu_n)\|_{L^2(X,X;\mu_n)}^2 = \frac{1}{\eta^2} \int_X \left\| x - T_{\mu_n}^{\text{JKO}_{\eta(\mathcal{E}_G+\mathcal{H})}((I+\eta S)_{\#}\mu_n)}(x) \right\|^2 d\mu_n(x)$$

$$= \frac{1}{\eta^2} W_2^2(\mu_n, \text{JKO}_{\eta(\mathcal{E}_G+\mathcal{H})}((I + \eta S)_{\#}\mu_n))$$

$$= \frac{1}{\eta^2} W_2^2(\mu_n, \mu_{n+1}).$$

On the other hand, it follows from (25) of the proof of Thm. 1 that

$$\min_{i=\overline{1,N}} W_2^2(\mu_n, \mu_{n+1}) = O(N^{-1}).$$

### A.6  Proof of Theorem 3

$H$ has uniformly bounded Hessian, by [43, Prop. 2.12], $\mathcal{E}_H$ is Wasserstein differentiable and $\nabla_W \mathcal{E}_H(\mu) = \nabla H$ for all $\mu \in \mathcal{P}_2(X)$. According to Lem. 8 and (18),

$$\partial_F^- \mathcal{F}(\mu_{n+1}) = \partial(\mathcal{E}_G + \mathcal{H})(\mu_{n+1}) - \nabla H \ni \frac{T_{\mu_{n+1}}^{\nu_{n+1}} - I}{\eta} - \nabla H. \tag{39}$$

We then have the following evaluations:

$$\text{dist}\left(0, \partial_F^- \mathcal{F}(\mu_{n+1})\right) = \inf_{\xi \in \partial_F^- \mathcal{F}(\mu_{n+1})} \|\xi\|_{L^2(X,X,\mu_{n+1})}$$

$$\leq \left\| \frac{T_{\mu_{n+1}}^{\nu_{n+1}} - I}{\eta} - \nabla H \right\|_{L^2(X,X,\mu_{n+1})}$$

$$= \left( \int_X \left\| \frac{T_{\mu_{n+1}}^{\nu_{n+1}}(x) - x}{\eta} - \nabla H(x) \right\|^2 d\mu_{n+1}(x) \right)^{\frac{1}{2}}$$

$$= \frac{1}{\eta} \left( \int_X \|T_{\mu_{n+1}}^{\nu_{n+1}}(x) - x - \eta \nabla H(x)\|^2 d\mu_{n+1}(x) \right)^{\frac{1}{2}}. \tag{40}$$

By transfer lemma 2,

$$\int_X \left\| T_{\mu_{n+1}}^{\nu_{n+1}}(x) - x - \eta \nabla H(x) \right\|^2 d\mu_{n+1}(x)$$

$$= \int_X \left\| T_{\mu_{n+1}}^{\nu_{n+1}}(x) - x - \eta \nabla H(x) \right\|^2 dT_{\nu_{n+1}}^{\mu_{n+1}} {}_\# \nu_{n+1}(x)$$

$$= \int_X \left\| T_{\mu_{n+1}}^{\nu_{n+1}} \circ T_{\nu_{n+1}}^{\mu_{n+1}}(x) - (I + \eta \nabla H) \circ T_{\nu_{n+1}}^{\mu_{n+1}}(x) \right\|^2 d\nu_{n+1}(x)$$

$$= \int_X \left\| x - (I + \eta \nabla H) \circ T_{\nu_{n+1}}^{\mu_{n+1}}(x) \right\|^2 d\nu_{n+1}(x). \tag{41}$$

On the other hand, by using the trivial identity

$$\nabla H = \frac{(I + \eta \nabla H) - I}{\eta}$$

we compute,

$$\int_X \|\nabla H(T_{\nu_{n+1}}^{\mu_n}(x)) - \nabla H(T_{\nu_{n+1}}^{\mu_{n+1}}(x))\|^2 d\nu_{n+1}(x)$$

$$= \int_X \left\| \frac{(I + \eta \nabla H) \circ T_{\nu_{n+1}}^{\mu_n}(x) - T_{\nu_{n+1}}^{\mu_n}(x)}{\eta} - \frac{(I + \eta \nabla H) \circ T_{\nu_{n+1}}^{\mu_{n+1}}(x) - T_{\nu_{n+1}}^{\mu_{n+1}}(x)}{\eta} \right\|^2 d\nu_{n+1}(x)$$

$$= \frac{1}{\eta^2} \int_X \left\| x - T_{\nu_{n+1}}^{\mu_n}(x) - (I + \eta \nabla H) \circ T_{\nu_{n+1}}^{\mu_{n+1}}(x) + T_{\nu_{n+1}}^{\mu_{n+1}}(x) \right\|^2 d\nu_{n+1}(x), \tag{42}$$

where the last equality uses $(I + \eta \nabla H) \circ T_{\nu_{n+1}}^{\mu_n} = I \ \nu_{n+1}$-a.e.

The Hessian of $H$ is bounded uniformly, $\nabla H$ is Lipschitz, let's say $\|\nabla H(x) - \nabla H(y)\| \leq L_H \|x-y\|$ for all $x, y \in X$. We continue evaluating (41) as follows

$$\int_X \|x - (I + \eta \nabla H) \circ T_{\nu_{n+1}}^{\mu_{n+1}}(x)\|^2 d\nu_{n+1}(x)$$

$$\leq \int_X \left( \|x - T_{\nu_{n+1}}^{\mu_n}(x) - (I + \eta \nabla H) \circ T_{\nu_{n+1}}^{\mu_{n+1}}(x) + T_{\nu_{n+1}}^{\mu_{n+1}}(x)\| + \|T_{\nu_{n+1}}^{\mu_n}(x) - T_{\nu_{n+1}}^{\mu_{n+1}}(x)\| \right)^2 d\nu_{n+1}(x)$$

$$\leq 2 \int_X \left\| x - T_{\nu_{n+1}}^{\mu_n}(x) - (I + \eta \nabla H) \circ T_{\nu_{n+1}}^{\mu_{n+1}}(x) + T_{\nu_{n+1}}^{\mu_{n+1}}(x) \right\|^2 d\nu_{n+1}(x)$$

$$+ 2 \int_X \|T_{\nu_{n+1}}^{\mu_n}(x) - T_{\nu_{n+1}}^{\mu_{n+1}}(x)\|^2 d\nu_{n+1}(x)$$

$$= 2\eta^2 \int_X \|\nabla H(T_{\nu_{n+1}}^{\mu_n}(x)) - \nabla H(T_{\nu_{n+1}}^{\mu_{n+1}}(x))\|^2 d\nu_{n+1}(x) + 2 \int_X \|T_{\nu_{n+1}}^{\mu_n}(x) - T_{\nu_{n+1}}^{\mu_{n+1}}(x)\|^2 d\nu_{n+1}(x)$$

$$\leq 2(\eta^2 L_H^2 + 1) \int_X \|T_{\nu_{n+1}}^{\mu_n}(x) - T_{\nu_{n+1}}^{\mu_{n+1}}(x)\|^2 d\nu_{n+1}(x) \tag{43}$$

where the third equality uses (42).

From (40), (41), and (43), we derive

$$\text{dist}\left(0, \partial_F^- \mathcal{F}(\mu_{n+1})\right) \le \frac{\sqrt{2(\eta^2 L_H^2 + 1)}}{\eta} \left(\int_X \|T_{\nu_{n+1}}^{\mu_n}(x) - T_{\nu_{n+1}}^{\mu_{n+1}}(x)\|^2 d\nu_{n+1}(x)\right)^{\frac{1}{2}}. \qquad (44)$$

On the other hand, by telescoping Lem. 1, we obtain

$$\sum_{n=1}^{\infty} \int_X \|T_{\nu_{n+1}}^{\mu_n}(x) - T_{\nu_{n+1}}^{\mu_{n+1}}(x)\|^2 d\nu_{n+1}(x) < +\infty.$$

Therefore,

$$\sum_{n=0}^{N-1} \text{dist}\left(0, \partial^F \mathcal{F}(\mu_{n+1})\right)$$

$$\le \frac{\sqrt{2(\eta^2 L_H^2 + 1)}}{\eta} \sum_{n=0}^{N-1} \left(\int_X \|T_{\nu_{n+1}}^{\mu_n}(x) - T_{\nu_{n+1}}^{\mu_{n+1}}(x)\|^2 d\nu_{n+1}(x)\right)^{\frac{1}{2}}$$

$$\le \frac{\sqrt{2(\eta^2 L_H^2 + 1)}}{\eta} \left(N \left(\sum_{n=0}^{N-1} \int_X \|T_{\nu_{n+1}}^{\mu_n}(x) - T_{\nu_{n+1}}^{\mu_{n+1}}(x)\|^2 d\nu_{n+1}(x)\right)\right)^{\frac{1}{2}}$$

$$\le \frac{\sqrt{2(\eta^2 L_H^2 + 1)N}}{\eta} \left(\sum_{n=0}^{+\infty} \int_X \|T_{\nu_{n+1}}^{\mu_n}(x) - T_{\nu_{n+1}}^{\mu_{n+1}}(x)\|^2 d\nu_{n+1}(x)\right)^{\frac{1}{2}}$$

We derive

$$\min_{n=\overline{1,N}} \text{dist}\left(0, \partial^F \mathcal{F}(\mu_n)\right) = O\left(\frac{1}{\sqrt{N}}\right).$$

### A.7 Proof of Theorem 4

**Convergence in terms of objective values**

Since $H \in C^2(X)$ whose Hessian is uniformly bounded, recall from (39) that

$$\partial_F^- \mathcal{F}(\mu_{n+1}) = \partial(\mathcal{E}_G + \mathscr{H})(\mu_{n+1}) - \nabla H \ni \frac{T_{\mu_{n+1}}^{\nu_{n+1}} - I}{\eta} - \nabla H.$$

Since $\mathcal{F}(\mu_0) - \mathcal{F}^* < r_0$ and the sequence $\{\mathcal{F}(\mu_n)\}_{n \in \mathbb{N}}$ is not increasing (Lem. 1), $\mathcal{F}(\mu_n) - \mathcal{F}^* < r_0$ for all $n \in \mathbb{N}$. Łojasiewicz condition implies

$$c(\mathcal{F}(\mu_{n+1}) - \mathcal{F}^*)^\theta \le \left\|\frac{T_{\mu_{n+1}}^{\nu_{n+1}} - I}{\eta} - \nabla H\right\|_{L^2(X,X,\mu_{n+1})}$$

$$= \left(\int_X \left\|\frac{T_{\mu_{n+1}}^{\nu_{n+1}}(x) - x}{\eta} - \nabla H(x)\right\|^2 d\mu_{n+1}(x)\right)^{\frac{1}{2}}$$

$$\le \frac{\sqrt{2(\eta^2 L_H^2 + 1)}}{\eta} \left(\int_X \|T_{\nu_{n+1}}^{\mu_n}(x) - T_{\nu_{n+1}}^{\mu_{n+1}}(x)\|^2 d\nu_{n+1}(x)\right)^{\frac{1}{2}} \qquad (45)$$

where the last inequality follows from (41) and (43) and $L_H$ is the Lipschitz constant of $\nabla H$. Combining with Lem. 1, we derive

$$c\left(\mathcal{F}(\mu_{n+1}) - \mathcal{F}^*\right)^\theta \le \frac{\sqrt{2(\eta^2 L_H^2 + 1)}}{\sqrt{\eta}} \left(\mathcal{F}(\mu_n) - \mathcal{F}(\mu_{n+1})\right)^{\frac{1}{2}}$$

or

$$\left(\mathcal{F}(\mu_{n+1}) - \mathcal{F}^*\right)^{2\theta} \le \frac{2(\eta^2 L_H^2 + 1)}{c^2 \eta} \left((\mathcal{F}(\mu_n) - \mathcal{F}^*) - (\mathcal{F}(\mu_{n+1}) - \mathcal{F}^*)\right). \qquad (46)$$

We then use the following lemma [71, Lem. 4].

**Lemma 11.** *Let $\{s_k\}_{k\in\mathbb{N}}$ be a nonincreasing and nonnegative real sequence. Assume that there exist $\alpha \geq 0$ and $\beta > 0$ such that for all sufficiently large $k$,*

$$s_{k+1}^\alpha \leq \beta(s_k - s_{k+1}). \tag{47}$$

*Then*

    *(i) if $\alpha = 0$, the sequence $\{s_k\}_{k\in\mathbb{N}}$ converges to $0$ in a finite number of steps;*

    *(ii) if $\alpha \in (0,1]$, the sequence $\{s_k\}_{k\in\mathbb{N}}$ converges linearly to $0$ with rate $\frac{\beta}{\beta+1}$;*

    *(iii) if $\alpha > 1$, the sequence $\{s_k\}_{k\in\mathbb{N}}$ converges sublinearly to $0$, i.e., there exists $\tau > 0$:*

$$s_k \leq \tau k^{\frac{-1}{\alpha-1}}$$

    *for sufficiently large $k$.*

Compared to [71, Lem. 4], we have dropped the assumption $s_k \to 0$ in Lem. 11 because this assumption is vacuous, i.e., it can be induced by (47) and nonnegativity of $\{s_k\}_{k\in\mathbb{N}}$.

We now apply Lem. 11 for $s_k = \mathcal{F}(\mu_k) - \mathcal{F}^*$ using (46) to derive the followings

    (i) if $\theta = 0$, $\mathcal{F}(\mu_n) - \mathcal{F}^*$ converges to $0$ in a finite number of steps;

    (ii) if $\theta \in (0, 1/2]$, $\mathcal{F}(\mu_n) - \mathcal{F}^*$ converges to $0$ linearly (exponentially fast) with rate

$$\mathcal{F}(\mu_n) - \mathcal{F}^* = O\left(\left(\frac{M}{M+1}\right)^n\right) \text{ where } M = \frac{2(\eta^2 L_H^2 + 1)}{c^2 \eta};$$

    (iii) if $\theta \in (1/2, 1)$, $\mathcal{F}(\mu_n) - \mathcal{F}^*$ converges sublinearly to $0$, i.e.,

$$\mathcal{F}(\mu_n) - \mathcal{F}^* = O\left(n^{-\frac{1}{2\theta-1}}\right).$$

## A.8   Proof of Theorem 5

**Cauchy sequence.**

By replacing $n := n - 1$ in (45) and rearranging

$$1 \leq \frac{\sqrt{2(\eta^2 L_H^2 + 1)}}{c\eta}\left(\int_X \|T_{\nu_n}^{\mu_{n-1}}(x) - T_{\nu_n}^{\mu_n}(x)\|^2 d\nu_n(x)\right)^{\frac{1}{2}} (\mathcal{F}(\mu_n) - \mathcal{F}^*)^{-\theta}. \tag{48}$$

It follows from Lem. 1 and (48) that

$$\int_X \|T_{\nu_{n+1}}^{\mu_n}(x) - T_{\nu_{n+1}}^{\mu_{n+1}}(x)\|^2 d\nu_{n+1}(x) \leq \eta(\mathcal{F}(\mu_n) - \mathcal{F}(\mu_{n+1}))$$

$$\leq \frac{\sqrt{2(\eta^2 L_H^2 + 1)}}{c}\left(\int_X \|T_{\nu_n}^{\mu_{n-1}}(x) - T_{\nu_n}^{\mu_n}(x)\|^2 d\nu_n(x)\right)^{\frac{1}{2}} (\mathcal{F}(\mu_n) - \mathcal{F}^*)^{-\theta} (\mathcal{F}(\mu_n) - \mathcal{F}(\mu_{n+1})). \tag{49}$$

Since the function $s : \mathbb{R}^+ \to \mathbb{R}$, $s(t) = t^{1-\theta}$ is concave if $\theta \in [0, 1)$, tangent inequality holds

$$s'(a)(a - b) \leq s(a) - s(b).$$

Note that $s'(t) = (1 - \theta)t^{-\theta}$, the above inequality further implies

$$(1 - \theta)(\mathcal{F}(\mu_n) - \mathcal{F}^*)^{-\theta}(\mathcal{F}(\mu_n) - \mathcal{F}(\mu_{n+1})) \leq (\mathcal{F}(\mu_n) - \mathcal{F}^*)^{1-\theta} - (\mathcal{F}(\mu_{n+1}) - \mathcal{F}^*)^{1-\theta}. \tag{50}$$

From (49) and (50)

$$\int_X \|T_{\nu_{n+1}}^{\mu_n}(x) - T_{\nu_{n+1}}^{\mu_{n+1}}(x)\|^2 d\nu_{n+1}(x) \leq \frac{\sqrt{2(\eta^2 L_H^2 + 1)}}{(1-\theta)c}\left(\int_X \|T_{\nu_n}^{\mu_{n-1}}(x) - T_{\nu_n}^{\mu_n}(x)\|^2 d\nu_n(x)\right)^{\frac{1}{2}}$$
$$\times \left[(\mathcal{F}(\mu_n) - \mathcal{F}^*)^{1-\theta} - (\mathcal{F}(\mu_{n+1}) - \mathcal{F}^*)^{1-\theta}\right]$$

or equivalently,

$$\frac{r_n}{\sqrt{r_{n-1}}} := \frac{\int_X \|T^{\mu_n}_{\nu_{n+1}}(x) - T^{\mu_{n+1}}_{\nu_{n+1}}(x)\|^2 d\nu_{n+1}(x)}{\left(\int_X \|T^{\mu_{n-1}}_{\nu_n}(x) - T^{\mu_n}_{\nu_n}(x)\|^2 d\nu_n(x)\right)^{\frac{1}{2}}}$$

$$\leq \frac{\sqrt{2(\eta^2 L_H^2 + 1)}}{(1-\theta)c} \left[(\mathcal{F}(\mu_n) - \mathcal{F}^*)^{1-\theta} - (\mathcal{F}(\mu_{n+1}) - \mathcal{F}^*)^{1-\theta}\right] \tag{51}$$

where $r_n := \int_X \|T^{\mu_n}_{\nu_{n+1}}(x) - T^{\mu_{n+1}}_{\nu_{n+1}}(x)\|^2 d\nu_{n+1}(x)$.

By telescoping (51) from $n = 1$ to $+\infty$ we obtain

$$\sum_{n=1}^{+\infty} \frac{r_n}{\sqrt{r_{n-1}}} < +\infty.$$

On the other hand

$$\frac{r_n}{\sqrt{r_{n-1}}} + \sqrt{r_{n-1}} \geq 2\sqrt{r_n}, \tag{52}$$

we derive $\sum_{n=0}^{+\infty} \sqrt{r_n} < +\infty$. From (22), (23) of the proof of Thm. 1, $r_n \geq W_2^2(\mu_n, \mu_{n+1})$, we obtain

$$\sum_{n=0}^{+\infty} W_2(\mu_n, \mu_{n+1}) < +\infty.$$

or, in other words, $\{\mu_n\}_{n\in\mathbb{N}}$ is a Cauchy sequence under Wasserstein topology. The Wasserstein space $(\mathcal{P}_2(X), W_2)$ is *complete* [5, Thm. 2.2], every Cauchy sequence is convergent, i.e., there exists $\mu^* \in \mathcal{P}_2(X)$ such that $\mu_n \xrightarrow{\text{Wass}} \mu^*$.

We prove that $\mu^*$ is actually an optimal solution of $\mathcal{F}$ by showing $\mathcal{F}(\mu^*) = \mathcal{F}^*$. Indeed, firstly, as $G$ and $H$ have quadratic growth, it holds

$$\mathcal{E}_G(\mu_n) \to \mathcal{E}_G(\mu^*), \mathcal{E}_H(\mu_n) \to \mathcal{E}_H(\mu^*).$$

On the other hand,

$$\mathcal{F}^* = \lim_{n\to\infty} \mathcal{F}(\mu_n) = \liminf_{n\to\infty} \mathcal{F}(\mu_n) = \liminf_{n\to\infty} \mathscr{H}(\mu_n) + \mathcal{E}_G(\mu^*) - \mathcal{E}_H(\mu^*)$$

$$\geq \mathscr{H}(\mu^*) + \mathcal{E}_G(\mu^*) - \mathcal{E}_H(\mu^*) = \mathcal{F}(\mu^*)$$

since $\mathscr{H}$ is l.s.c. The equality has to occur, i.e., $\mathcal{F}^* = \mathcal{F}(\mu^*)$, due to the optimality of $\mathcal{F}^*$.

**Convergence rate of $\{\mu_n\}_{n\in\mathbb{N}}$.**

*(i) If $\theta = 0$*

From item (i) of Thm. 4, there exists $n_0 \in \mathbb{N}$ such that $\mathcal{F}(\mu_n) = \mathcal{F}^*$ for all $n \geq n_0$. It then follows from (24) that $\mu_{n_0} = \mu_{n_0+1} = \mu_{n_0+2} = \ldots$, which further implies that $\mu_n = \mu^*$ for all $n \geq n_0$.

*(ii) If $\theta \in (0, 1/2]$*

Let $s_i = \sum_{n=i}^{\infty} \sqrt{r_n}$. We have

$$s_i \geq \sum_{n=i}^{\infty} W_2(\mu_n, \mu_{n+1}) \geq W_2(\mu_i, \mu^*) \tag{53}$$

where the last inequality uses triangle inequality

$$W_2(\mu_i, \mu^*) \leq \sum_{n=i}^{N-1} W_2(\mu_n, \mu_{n+1}) + W_2(\mu_N, \mu^*)$$

and lets $N \to \infty$ with a notice that $\mu_N \xrightarrow{\text{Wass}} \mu^*$.

From (51) and (52),

$$2\sqrt{r_n} \leq \sqrt{r_{n-1}} + \frac{\sqrt{2(\eta^2 L_H^2 + 1)}}{(1-\theta)c} \left[(\mathcal{F}(\mu_n) - \mathcal{F}^*)^{1-\theta} - (\mathcal{F}(\mu_{n+1}) - \mathcal{F}^*)^{1-\theta}\right]. \tag{54}$$

Telescope (54) for $n = i$ to $+\infty$,

$$s_i \leq \sqrt{r_{i-1}} + \frac{\sqrt{2(\eta^2 L_H^2 + 1)}}{(1-\theta)c}(\mathcal{F}(\mu_i) - \mathcal{F}^*)^{1-\theta} \leq \sqrt{r_{i-1}} + \frac{(2(\eta^2 L_H^2 + 1))^{\frac{1}{2\theta}}}{(1-\theta)\eta^{\frac{1-\theta}{\theta}}c^{\frac{1}{\theta}}} r_{i-1}^{\frac{1-\theta}{2\theta}}. \quad (55)$$

where the last inequality uses (45). Since $r_i \to 0$ as $i \to \infty$, $r_i < 1$ for $i$ sufficiently large. It follows from (55) that: for $i$ sufficiently large

$$s_i \leq M\sqrt{r_{i-1}} = M(s_{i-1} - s_i)$$

where

$$M = 1 + \frac{(2(\eta^2 L_H^2 + 1))^{\frac{1}{2\theta}}}{(1-\theta)\eta^{\frac{1-\theta}{\theta}}c^{\frac{1}{\theta}}}. \quad (56)$$

Rewriting as $s_i \leq \frac{M}{M+1}s_{i-1}$, we derive $W_2(\mu_i, \mu^*) = O\left(\left(\frac{M}{M+1}\right)^i\right)$.

*(iii) If $\theta \in (1/2, 1)$*

(55) implies: for all $i$ sufficiently large,

$$s_i \leq Mr_{i-1}^{\frac{1-\theta}{2\theta}} = M(s_{i-1} - s_i)^{\frac{1-\theta}{\theta}}$$

where $M$ is the same as in (56).

Applying Lem. 11(iii), $s_i = O\left(i^{-\frac{1-\theta}{2\theta-1}}\right)$, which implies (by (53)) $W_2(\mu_i, \mu^*) = O\left(i^{-\frac{1-\theta}{2\theta-1}}\right)$.

# B    Implementation of FB Euler

Following a similar approach to the semi FB Euler's implementation outlined in Alg. 1, we present a practical implementation of FB Euler for the sampling context in Alg. 2.

---
**Algorithm 2** FB Euler for sampling
---
**Input:** Initial measure $\mu_0 \in \mathcal{P}_{2,\mathrm{abs}}(X)$, discretization stepsize $\eta > 0$, number of steps $K > 0$, batch size $B$.
**for** $k = 1$ to $K$ **do**
    **for** $i = 1, 2, \ldots$ **do**
        Draw a batch of samples $Z \sim \mu_0$ of size $B$;
        $\Xi \leftarrow (I - \eta\nabla F) \circ \nabla_x\psi_{\theta_k} \circ (I - \eta\nabla F) \circ \nabla_x\psi_{\theta_{k-1}} \circ \ldots \circ (I - \eta\nabla F)(Z)$;
        $\widehat{W_2^2} \leftarrow \frac{1}{B}\sum_{\xi \in \Xi}\|\nabla_x\psi_\theta(\xi) - \xi\|^2$;
        $\widehat{\Delta\mathscr{H}} \leftarrow -\frac{1}{B}\sum_{\xi \in \Xi}\log\det\nabla_x^2\psi_\theta(\xi)$.
        $\widehat{\mathcal{L}} \leftarrow \frac{1}{2\eta}\widehat{W_2^2} + \widehat{\Delta\mathscr{H}}$.
        Apply an optimization step (e.g., Adam) over $\theta$ using $\nabla_\theta\widehat{\mathcal{L}}$.
    **end for**
    $\theta_{k+1} \leftarrow \theta$.
**end for**
---

# C    Numerical illustrations

We perform numerical experiments in a high-performance computing cluster with GPU support. We use Python version 3.8.0. We allocate 8G memory for the experiments. The total running time for all experiments is a couple of hours. Our implementation is based on the code of [53] (MIT license) with the DenseICNN architecture [41].

## C.1 Gaussian mixture

Consider a target Gaussian mixture of the following form:

$$\pi(x) \propto \exp(-F(x)) := \sum_{i=1}^{K} \pi_i \exp\left(-\frac{\|x - x_i\|^2}{\sigma^2}\right).$$

We write

$$
\begin{aligned}
F(x) &= -\log\left(\sum_{i=1}^{K} \pi_i \exp\left(-\frac{\|x - x_i\|^2}{\sigma^2}\right)\right) \\
&= -\log\left(\sum_{i=1}^{K} \pi_i \exp\left(-\frac{\|x\|^2 + \|x_i\|^2 - 2\langle x, x_i \rangle}{\sigma^2}\right)\right) \\
&= -\log\left(\sum_{i=1}^{K} \pi_i \exp\left(-\frac{\|x\|^2}{\sigma^2}\right) \times \exp\left(-\frac{\|x_i\|^2 - 2\langle x, x_i \rangle}{\sigma^2}\right)\right) \\
&= \frac{\|x\|^2}{\sigma^2} \underbrace{- \log\left(\sum_{i=1}^{K} \pi_i \exp\left(-\frac{\|x_i\|^2 - 2\langle x, x_i \rangle}{\sigma^2}\right)\right)}_{\text{convex}}
\end{aligned}
$$

which is DC. Note that the convexity of the second component is thanks to (a) log-sum-exp is convex and (b) the composite of a convex function and an affine function is convex.

**Experiment details** We set $K = 5$ and randomly generate $x_1, x_2, \ldots, x_5 \in \mathbb{R}^2$. We set $\sigma = 1$. The initial distribution is $\mu_0 = \mathcal{N}(0, 16I)$. We use $\eta = 0.1$ for both FB Euler and semi FB Euler. We train both algorithms for 40 iterations using Adam optimizer with a batch size of 512 in which the first 20 iterations use a learning rate of $5 \times 10^{-3}$ while the latter 20 iterations use $2 \times 10^{-3}$. For the baseline ULA, we run 10000 chains in parallel for 4000 iterations with a learning rate of $10^{-3}$.

We run the above experiment 5 times where $x_1, x_2, \ldots, x_5$ are randomly generated each time. Fig. 1 (b) reports the KL divergence along the training process where the mean curves are in bold and individual curves are displayed in a faded style. The final KL divergence (averaged across 5 runs) of the ULA is reported as a horizontal line.

## C.2 Distance-to-set prior

Let $\pi$ be the original prior, $\Theta$ be the constraint set that we want to impose, and the distance-to-set prior [61] is defined by, for some $\rho > 0$,

$$\tilde{\pi}(\theta) \propto \pi(\theta) \exp\left(-\frac{\rho}{2} d(\theta, \Theta)^2\right),$$

that penalize exponentially $\theta$ deviating from the constraint set.

Given data $y$, using the this distance-to-set prior, the posterior reads

$$\bar{\pi}(\theta|y) \propto L(\theta|y)\pi(\theta) \exp\left(-\frac{\rho}{2} d(\theta, \Theta)^2\right).$$

where $L(\theta|y)$ is the likelihood.

The structure of $\bar{\pi}(\theta|y)$ depends on three separate components: the original prior, the likelihood, and the constraint set $\Theta$. In the ideal case, $\pi(\theta)$ and $L(\theta|y)$ are given in nice forms (e.g., log-concave), and $\Theta$ is a convex set. As a fact, if $\Theta$ is a convex set, $\theta \mapsto d(\theta, \Theta)^2$ is convex, making the whole posterior log-concave. If $\Theta$ is additionally closed, $d(\theta, \Theta)^2$ is L-smooth.

However, whenever $\Theta$ is nonconvex, the function $\theta \mapsto d(\theta, \Theta)^2$ is not continuously differentiable. This is induced by the Motzkin-Bunt theorem [16, Thm. 9.2.5] asserting that any Chebyshev set (a set $S \subset X$ is called Chebyshev if every point in $X$ has a unique nearest point in $S$) has to be closed and convex.

On the other hand, $d(\theta, \Theta)^2$ is always DC regardless the geometric structure of $\Theta$:

$$
\begin{aligned}
d(\theta, \Theta)^2 &= \inf_{x \in \Theta} \|\theta - x\|^2 \\
&= \inf_{x \in \Theta} \left( \|\theta\|^2 + \|x\|^2 - 2\langle x, \theta \rangle \right) \\
&= \|\theta\|^2 + \inf_{x \in \Theta} \left( \|x\|^2 - 2\langle x, \theta \rangle \right) \\
&= \|\theta\|^2 - \sup_{x \in \Theta} \left( -\|x\|^2 + 2\langle x, \theta \rangle \right).
\end{aligned}
\tag{57}
$$

Note that the supremum of an *arbitrary* family of affine functions is convex.

Therefore, the log-DC structure of the whole posterior only depends on whether the original prior and the likelihood are log-DC, which is likely to be the case.

**Distance-to-set prior relaxed von Mises-Fisher** In directional statistics, the von Mises-Fisher distribution is a distribution over unit-length vectors (unit sphere). It can be described as a restriction of a Gaussian distribution in a sphere. By using the distance-to-set prior, we can relax the spherical constraint as

$$
\bar{\pi}(\theta) \propto \exp(-F(\theta)) := \exp\left( -\kappa \frac{(\theta - \mu)^\top (\theta - \mu)}{2} \right) \times \exp\left( -\frac{\rho}{2} \operatorname{dist}(\theta, S)^2 \right)
$$

where $S$ denote the unit sphere in some $\mathbb{R}^d$ space. By the DC structure (57) of the distance function, $F$ is DC with the following composition

$$
\begin{aligned}
F(\theta) &= \left( \kappa \frac{\|\theta - \mu\|^2}{2} + \frac{\rho}{2} \|\theta\|^2 \right) - \frac{\rho}{2} \sup_{x \in S} \left( -\|x\|^2 + 2\langle x, \theta \rangle \right) \\
&:= G(\theta) - H(\theta).
\end{aligned}
$$

We also note that $H$ is not continuously differentiable because $S$ is nonconvex. Furthermore, $\rho \operatorname{proj}_S(\theta) \in \partial H(\theta)$ where $\operatorname{proj}_S(\theta)$ is the projection of $\theta$ onto $S$, which can be computed explicitly in this case.

**Experiment details** We consider a unit circle in $\mathbb{R}^2$ with centre $\mu = (1, 1.5)$. We set $\kappa = 1$ and $\rho = 100$. The initial distribution is $\mu_0 = \mathcal{N}(0, 16I)$. We use $\eta = 0.1$ for both FB Euler and semi FB Euler. We train both algorithms for 40 iterations using Adam optimizer with a batch size of 512 in which the first 20 iterations use a learning rate of $5 \times 10^{-3}$ while the latter 20 iterations use $2 \times 10^{-3}$.

