# OpenReview forum: "Non-geodesically-convex optimization in the Wasserstein space"
_NeurIPS.cc/2024/Conference — NeurIPS 2024 poster_

### Official Review · Reviewer_7Mmt · 2024-06-19

**Soundness:** 4
**Presentation:** 3
**Contribution:** 1
**Rating:** 3
**Confidence:** 4

**Summary:**

This paper introduces and analyzes an optimization scheme, termed "semi-FB Euler", for a particular minimization problem over the Wasserstein space $P_2(R^d)$: $\min_\mu \mathcal{F}(\mu) = \int (G-H) d\mu + \mathcal{H}(\mu)$, where $G$ and $H$ are convex functions over $R^d$ and $\mathcal{H}$ is a functional that is convex over generalized geodesics. Namely, "semi-FB Euler" is defined by equation (4) of the paper: starting from some $\mu_0$,
$$\begin{cases}
\nu\_{n+1} = (I + \gamma \nabla H)\_{\sharp} \mu_n, \\\\
\mu\_{n+1} \in \arg\min\_{P\_2(R^d)} \int G d\mu + \mathcal{H}(\mu) + \frac{1}{2\gamma} W\_2^2(\cdot, \nu\_{n+1}).
\end{cases}$$

(The second step, called a JKO step, is assumed solvable for the theoretical parts of the paper.)
Convergence to critical points of $\mathcal{F}$ is established under mild regularity assumptions, both asymptotically (Theorem 1) and with non-asymptotic rates (Theorems 2, 3). Under a Lojasiewicz inequality assumption, convergence to a minimizer is established with non-asymptotic rates (Theorems 4, 5).

Numerical illustrations with synthetic data are provided, in which an input-convex neural network is used to implement the JKO step.

**Strengths:**

This work provides mathematically rigorous grounding for an infinite-dimensional optimization scheme, by working out explicit regularity conditions that ensure well-definedness and asymptotic convergence. The precision and attention to rigor are appreciable.

The paper is clearly written and well-structured. It also provides an appropriate level of background information on the Wasserstein geometry.

Compared to the closely related work "The Wasserstein proximal gradient algorithm" ([56] Salim, Korba, Luise, 2020), numerical experiments using input-convex neural networks (following [44] Morkov et al., 2021) illustrate that the JKO step can indeed be implemented in general, although computational costs are not discussed.

**Weaknesses:**

The proposed "semi-FB Euler" corresponds to a forward-backward splitting scheme of $\mathcal{F}$ into $\mathcal{F}_1(\mu) = \int (-H) d\mu$ and $\mathcal{F}_2(\mu) = \int G d\mu + \mathcal{H}(\mu)$ in the Wasserstein optimization geometry. (Incidentally, the name "semi-FB Euler" can be misleading, as it is really a forward-backward scheme.)
So the setting considered in this paper is very similar to that of "The Wasserstein proximal gradient algorithm" ([56] Salim, Korba, Luise, 2020), which corresponds to the case where $H$ is concave and $G=0$. In fact, it seems that the only way that convexity of $G$ comes into play in the proofs, is to ensure convexity of $\mathcal{F}_2$ over generalized geodesics. (Please do correct me if this is incorrect.)

With this in mind, the contributions of this paper are to study what happens when $-H$ is concave in $\mathcal{F}_1(\mu) = \int (-H) d\mu$, as opposed to convex in [56]. It seems that concavity of $-H$ is used for:
1. the existence of a Borel measurable selector $S$ for the subdifferential $\partial H$ (section A.1), but it is not essential to the paper;
2. the descent lemma (Lemma 1);
3. quadratic growth of $\|S\|^2$ under quadratic growth of $H$ (Lemma 4 and proof of Thm 1 line 706);
4. it seems that convexity of $H$ is implicity used in the very last step of the proof of Thm 1 (line 735). (Incidentally, please make the last step of the proof of Thm 1 more explicit.)

In summary, compared to the setting of [56], which corresponds to $-H$ being smooth and convex, this work shows that if $-H$ is concave, then the forward-backward algorithm (with a good choice of the selector $S$ when $H$ is non-differentiable) converges for any choice of step-size $\gamma$.

I find this theoretical contribution interesting, but I am not sure if it is sufficiently significant. Besides, there is also the weakness, acknowledged by the authors (line 362) that the proposed algorithm relies on a JKO step which is computationally expensive. Hence my overall rating for this submission.

**Questions:**

- Please address the points in "Weaknesses".
- The items 1.-4. in "Weaknesses can also be guaranteed if, instead of $-H$ concave, one assumes $H$ is smooth (which is Assumption 5) and $\gamma$ is small enough (see [56, Thm 5] for item 2 and [56, Lemma 2] for item 4). Could your results be extended to the case where $H$ is smooth instead of convex? If yes, then this could be another way to show theoretical guarantees for forward-backward schemes outside of the convex case (in the spirit of your conclusion, line 393).
- What are the "interesting avenues for future work" mentioned in the conclusion?

**Limitations:**

The authors adequately addressed the limitations.

---

> ### Author Rebuttal · Authors · 2024-08-06
>
> We thank the reviewer for the positive comments on our work. We hope the reply below can highlight our contribution.
>
> **Strengths**
>
> ***"Computational cost..."***: as the answer to reviewer 7EW7, we will discuss the computational cost of the JKO in the revised version.
>
> **Weaknesses**
>
> ***``the setting considered in this paper is very similar to"***: our problem setting is much more general than that of ([56], Salim, Korba, Luise - NeurIPS2020). Having some concavity looks simple yet the whole objective structure becomes much more expressive. The expressivity can be acknowledged from the following result: **any continuous function can be approximated by a sequence of DC functions over a compact, convex domain [Bačák11]**.
>
> *[Bačák11] Bačák, M., & Borwein, J. M. (2011). On difference convexity of locally Lipschitz functions. Optimization, 60(8-9), 961-978.*
>
> By that, we mean that the structure is semantically simple but powerful mathematically.
>
> Saying that the convexity in [56] is replaced by the concavity in our work does not entirely reflect our framework, hence the contribution. We should look at the whole landscape of the objective function (potential energy + internal/interaction energy), and it would be more complete to say that our work replaces the geodesic convexity in [56] by geodesic difference-of-convexity.
>
> Apart from the theoretical contribution mentioned by the reviewer, we think there are two more elegant contributions:
>
>   (1) The use of the DC function for the potential energy already allows us to tackle sampling from log-DC density -- which is a large class of densities (instead of log-concave "and" log-smooth density as in ([56] Salim, Korba, Luise, 2020)).
>
>    (2) The way we separate $\mathcal{E}_G$ into the JKO part and leave $\mathcal{E}_H$ for the push-forward is non-trivial and is already a contribution (in our opinion). It differs from the common mindset of applying forward to the potential and backward to the internal energy. Our approach makes both steps tractable, i.e., the optimal map of the forward is computable and the optimal map of the backward is characterizable. Moreover, if we started from [56] with the old mindset and with a mere curiosity of what happens if we replace convex $H$ with concave $-H$, we would not do so in the first place and rather abandon the idea because there are not so many ``log-convex" distributions out there.
>
> For these reasons, we also would like to retain the name "semi-FB Euler" to address its non-triviality and distinguish it from [56].
>
> In terms of theory, the use of the concave $-H$ is correctly mentioned by the reviewer. However, we think the existence of $S$ is actually essential because it allows us to work with non-differentiable $H$ (as in Theorem 2). Without this concavity, we have to use more sophisticated subdifferential notions for $H$ like Clarke or Frechet [Clarke90] in the non-differentiability realm. These notions are difficult to work with, and they can be ill-posed (the subdifferential set is empty).
>
> *[Clarke90] Clarke, F. H. (1990). Optimization and nonsmooth analysis. Society for Industrial and Applied Mathematics.*
>
>
> ***``Please make the last step of the proof of Thm 1 more explicit"***: the equation between lines 734 and 735: $\gamma^{-1}(T_{\mu_*}^{\nu_*} - I) \in \partial (\mathcal{E}_G + \mathscr{H})(\mu^*)$.
>
> By the definition of $\nu^*$ in line 678, $\nu^* := (I+\gamma S)$#$\mu^*$, and since $S$ is the gradient field of a convex function $H$, that push-forward is optimal, i.e., $T_{\mu^*}^{\nu^*} = I + \gamma S$. Plug this in the above formula, we get $S \in \partial(\mathcal{E}_G + \mathscr{H})(\mu^*)$. On the other hand, since $H$ is convex, $S \in \partial \mathcal{E}_H(\mu^*)$ [4, Proposition 4.13].  We will add these explanations in the revised version.
>
>
> About the computational challenge of the JKO, please refer to our answer to reviewer 7EW7.
>
>
> **Questions**
>
> ***``Could your results be extended ..."***: We think our results are totally applicable for $L$-smooth potentials with small $\gamma$ as suggested. Note that, any $L$-smooth function is a DC function (see our paragraph **Why difference-of-convex structure**).  Therefore, while the suggestion is nice, it does not extend our work any further. The true extension could be that instead of geodesically-concave potential $-\mathcal{E}_H(\mu)$, we can consider a more general geodesic concavity not necessarily given in the form of potential energy.
>
> ***What are the "interesting avenues for future work"***: Those include, for example, the extension mentioned in the previous answer. Moreover, we can also investigate non-geodesically-convex problems in different Wasserstein spaces (either with different base space than $\mathbb{R}^d$, or different cost functions instead of squared Euclidean distance). For example, in [Section 5, Lambert2022], in the case of mixture of Gaussian Variational Inference, the base space for the Wasserstein space is $BW(\mathbb{R}^d)$ (Bures–Wasserstein space) instead of $\mathbb{R}^d$, i.e., we are minimizing over $\mathcal{P}_2(BW(\mathbb{R}^d))$ (Wasserstein space over Bures–Wasserstein space). This adds one layer of complexity and, as a consequence, convexity is lost in that space and, quoting from page 9 [Lambert2022], ``we lose many of the theoretical guarantees". We can extend the discussion when we have one page extra for the final version.
>
> *[Lambert2022] Lambert, M., Chewi, S., Bach, F., Bonnabel, S., & Rigollet, P. (2022). Variational inference via Wasserstein gradient flows. Advances in Neural Information Processing Systems, 35, 14434-14447.*

---

> ### Comment · Reviewer_7Mmt · 2024-08-08
>
> Thank you for clarifying what your contributions are. However, this clarification makes the significance of this work less convincing than I initially thought.
>
> The title of the submission "Non-geodesically-convex optimization in the Wasserstein space" indicates that 1. your audience is one that is interested in optimization questions; 2. you consider objective functionals which are non-convex, and the go-to condition in the non-convex optimization literature is smoothness. However your submission is actually about Difference-of-Convex optimization in the Wasserstein space, which is quite different, both in its application cases and in the kind of theory involved for the analysis.
>
> I understand the general appeal of the DC assumption, given the approximation result you cited ("any continuous function $f$ can be approximated by a sequence of DC functions $g-h$ over a compact, convex domain"). But a) this approximation result does not say how to choose the splitting $g-h$ in a usable way for optimization [please correct me if incorrect], and besides b) you do not prove such an approximation result for the Wasserstein space.
>
> Another argument in favor of studying DC functions on the Wasserstein space, instead of smooth ones, might be that your analysis holds under quite few assumptions. But, particularly in a venue such as NeurIPS, a theoretical analysis only has value when it applies (or there are reasons to think it will apply in the future) to (i) an algorithm _and_ (ii) a setting of practical interest. Now
> - (i) after reading your answer to Reviewer 7EW7, it is still unclear to me that the algorithm you analyze (using a proximal step in Wasserstein space) has hopes of becoming efficiently implementable. If, as you say, methods proposed 2 years ago allow for faster computation, then it is up to you to implement them.
> - (ii) The example setting you promoted most, namely sampling from a log-DC measure, deserves more discussion. Sampling problems of this form typically arise in Bayesian inference, where the target measure is a posterior likelihood which can be quite involved. It is not at all clear how to choose a splitting of it (or rather of its log) in a tractable way.
>
> For these reasons, I downgrade my rating from 4 to 3. Nonetheless I remain open to further discussion. My recommendation to the authors would be to exploit their (technically interesting!) analysis into a full-fledged study of DC optimization in the Wasserstein space. (I realize that this is not easy to incorporate into the current submission unfortunately.)
>
> PS1: I understand that your analysis can be extended to the case where the objective $\mathcal{F}$ is of the form $\tilde{\mathcal{G}} - \tilde{\mathcal{H}}$ where $\tilde{\mathcal{G}}, \tilde{\mathcal{H}}$ are general geodesically convex functionals, instead of $\tilde{\mathcal{G}}$ being linear ($\tilde{\mathcal{G}} = \mathcal{E}_{-H}$ and $\tilde{\mathcal{H}}= \mathcal{H} + \mathcal{E}_G$ in the paper). My comments above implicitly assume this extension.
>
> PS2, regarding the name semi-FB Euler: I do not have a strong opinion on the matter, but I would like to point out that choosing such a name does not contribute at all to convey the non-triviality of the method, and that this name would be quite awkward in the context of the natural extension mentioned in PS1 (with general $\tilde{\mathcal{G}} - \tilde{\mathcal{H}}$).

---

> > ### Author Response · Authors · 2024-08-09
> >
> > We thank the reviewer for the response.
> >
> > ***The title of the submission "Non-geodesically-convex optimization in the Wasserstein space" indicates...***
> >
> > We agree that smoothness (L-smooth) is the go-to condition when there is no convexity. As we understand, the reviewer advocates for this class of L-smooth nonconvex functions. We also assume we are talking about the nonconvexity of $F$ in the potential part, $\mathcal{E}_F$ (inducing the nonconvexity along Wasserstein geodesics). We emphasise is that the class of L-smooth  functions is just a subclass of DC functions thanks to the well-known result: if $F$ is L-smooth, then it admits the following DC decompositions: (1)
> > $F(x) = (F(x) + (\eta/2) \Vert x \Vert^2) - (\eta/2) \Vert x \Vert^2$
> > or (2)
> > $F(x) = (\eta/2) \Vert x \Vert^2 - ((\eta/2) \Vert x \Vert^2 - F(x))$
> > whenever $\eta \geq L$.
> >
> > In other words, our results readily apply to the class of $L$-smooth functions raised by the reviewer. Nevertheless, we can explicitly mention in the abstract that the nonconvex structure we are studying is DC.
> >
> > ***given the approximation result you cited...***
> >
> > The result we cited advocates the richness of the class of DC functions since it is a universal approximator for any continuous function over a compact region.
> >
> > Obtaining explicit DC decomposition for a given function has been studied thoroughly in the DC programming literature. It is known that many objective functions have natural DC decomposition (we are talking about the $F$ part in the potential) (see, e.g., [Nouiehed19, Lethi05]). We can add a paragraph to discuss the explicit decomposition in the revised version of the paper. Note that, in Appendix C.1 and C.2 in our submission, we already provided two concrete examples of the DC decomposition for the log of the Gaussian mixture and also von Mises-Fisher with distance-to-set prior, demonstrating that it can be done in practice. We can point out DC structures of other types of non-log-concave densities, including the posterior, as suggested.
> >
> > *[Nouiehed19] Nouiehed, M., Pang, J. S., \& Razaviyayn, M. (2019). On the pervasiveness of difference-convexity in optimization and statistics. Mathematical Programming, 174(1), 195-222.*
> >
> > *[Lethi05] H. A. Le Thi, \& T. Pham Dinh (2005). The DC (difference of convex functions) programming and DCA revisited with DC models of real world nonconvex optimization problems. Annals of operations research, 133, 23-46.*
> >
> > ***Another argument in favor of studying DC functions on the Wasserstein space, instead of smooth ones, might be that your analysis holds under quite a few assumptions***: This is indeed why we focus on DC class (containing L-smooth class), and we will make this advantage even more clear in the revised version. We already made an attempt to explain that in the paragraph **Why difference-of-convex structure** and will make it more pronounced.
> >
> > ***(i) It is still unclear to me that the algorithm you analyze (using a proximal step in Wasserstein space) has hopes of becoming efficiently implementable***: our work is mainly theoretical, in the same vein as in ([56], Salim, Korba, Luise - NeurIPS2020).
> >
> > ***(ii) The example setting you promoted most, namely sampling from a log-DC measure, deserves more discussion***: It is possible to discuss the DC structure of posterior distributions more extensively in the revised version. The splitting of the posterior depends on two terms: log-likelihood and log-prior. If we assume that the likelihood is nice enough, i.e., log-L-smooth, it is already DC with known splitting as above (of course, other structures can also be considered). The log-prior is normally not smooth to model sparsity or low rank (LASSO, group LASSO, SCAD, Capped-$\ell_1$, PiL, etc). Most of them have some explicit DC structure [Lethi15].
> >
> > *[Lethi15] Lethi, H. A., Dinh, T. P., Le, H. M., \& Vo, X. T. (2015). DC approximation approaches for sparse optimization. European Journal of Operational Research, 244(1), 26-46.*
> >
> >
> > ***My recommendation to the authors would be to exploit their (technically interesting!) analysis into a full-fledged study of DC optimization in the Wasserstein space***: this is a promising future research direction. However, our current paper is already technically large (26 pages or so), and integrating all of this into the current version makes it overly heavy. Moreover, we did not study the suggested class initially because we did not find any practical problems with the concave part given in a more general form than the potential energy.

---

> > > ### Comment · Reviewer_7Mmt · 2024-08-11
> > >
> > > Thank you for the detailed response!
> > > - It is indeed preferable to clarify from the outset (abstract or title) that the structure you are considering is DC, in my opinion.
> > > - I would recommend to pick a side clearly: either focus on sampling applications, with $\mathcal{H}$ being the entropy, or implement the extension to all Wasserstein-geodesically-DC functionals. The presentation in the submission is somewhat muddled, as general Wasserstein-geodesically-convex $\mathcal{H}$ is considered but all the examples and illustrations are for the entropy.
> > > - I was not aware of the fact that some "natural" Bayesian inference problems have a DC (with explicit splitting) but non-smooth structure, thank you for pointing this out. This fact should be promoted and discussed much more, in my opinion, as it shows that for certain settings of practical interest, DC-type considerations are in some sense unavoidable. This is much more convincing to me than the abstract argument that "more functions satisfy DC".
> > > - I do not find the theoretical contributions of your work sufficient to bypass the issue of practical performance. The implementability of the JKO step is indeed a difficult question. Perhaps you could motivate the practicality of your method by empirically comparing your method vs. traditional sampling methods on target measures which are DC but non-smooth?
> > >
> > > Looking forward to seeing your thoughts on these suggestions!

---

> > > > ### Author Response · Authors · 2024-08-12
> > > >
> > > > We thank the reviewer for the response.
> > > >
> > > >
> > > > ***$\bullet$ It is indeed preferable to clarify from the outset***: we will clarify in the abstract that we consider the DC structure.
> > > >
> > > > ***$\bullet$ I would recommend to pick a side clearly***: we would pick the general side that considers all geodesically convex regularizers because (1) we want to cover the case of Maximum Mean Discrepancy that has value in the context of deep learning theory, (2) our work is more about optimization in the Wasserstein space, motivated by sampling but should not be restricted to that area.
> > > >
> > > > About the implementation, apart from the negative entropy regularizer, we can discuss the following regularizers given in the following forms: (a) the interaction energy, (b) another (non-smooth) potential, and (c) general internal energy. We also note that the negative entropy considered in our paper is (probably) already the trickiest regularizer. It requires computing $E_{x \sim \rho_t}(\log(\rho_t(x)))$. Although samples from $\rho_t$ can be simulated, we cannot use the vanilla idea of Monte Carlo approximation because the function inside the expectation involves the probability density of $\rho_t$, which is unknown. The key technique making this possible is the change of entropy formula, saying that the change in the entropy can be computed via the determinant of the Jacobian of the transformation [Mokrov21] (line 828 in our work). For other types of regularizers, such as interaction or potential (or their summation), the implementation is straightforward (via Monte Carlo) since the expectation no longer involves the probability density. It is also worth noting that: in the case of the potential regularizer, the JKO operator reduces to the proximal operator in the Euclidean space [Wibisono18], and it is known that many proximal operators in the Euclidean space have closed-form solutions [Parikh14].
> > > >
> > > > On the other hand, the change of entropy formula can be naturally extended to the case of general internal energy $E_{x \sim \rho_t}(f(\rho_t))$ thanks to the change of variable formula [Lemmas 3.2, 3.4, A-Melis22]. This is pretty much in the same spirit as normalizing flows, another active ML research area.
> > > >
> > > > *[Mokrov21] Mokrov, P., Korotin, A., Li, L., Genevay, A., Solomon, J. M., \& Burnaev, E. (2021). Large-scale Wasserstein gradient flows. Advances in Neural Information Processing Systems, 34, 15243-15256.*
> > > >
> > > > *[A-Melis22] Alvarez-Melis, D., Schiff, Y., \& Mroueh, Y. (2022). Optimizing functionals on the space of probabilities with input convex neural networks. Transactions on Machine Learning Research.*
> > > >
> > > > *[Wibisono18] Wibisono, A. (2018). Sampling as optimization in the space of measures: The Langevin dynamics as a composite optimization problem. In Conference on Learning Theory (pp. 2093-3027). PMLR.*
> > > >
> > > > *[Parikh14] Parikh, N., \& Boyd, S. (2014). Proximal algorithms. Foundations and trends® in Optimization, 1(3), 127-239.*
> > > >
> > > >
> > > > ***$\bullet$ This fact should be promoted and discussed much more***: We will discuss more DC splittings for Bayesian posterior sampling as suggested.
> > > >
> > > > ***$\bullet$ Perhaps you could motivate the practicality of your method by empirically comparing your method vs. traditional sampling methods on target measures which are DC but non-smooth?***: we can compare with traditional sampling methods like ULA in the revised version. We agree with the reviewer's concern regarding the JKO scalability, as we also already explicitly admitted it in the main text of our submission (Line 362). Our work was mainly inspired by [36, Salim, Korba, Luise - NeurIPS2020], another purely theoretical work. We believe theoretical works have their own merits in the long run as they provide insight and connections that can guide practical research. The JKO operator, not only studied in the ML community but primarily by the PDE community, has demonstrated that purely theoretical objects can have far-reaching impacts, with the paper by Jordan, Kinderlehrer, and Otto in 1998 being seminal. Notice that even though our work is theoretical in nature, we have provided two concrete examples of DC decomposition, demonstrating that our approach can be implemented in practice.

---

> > > > > ### Comment · Reviewer_7Mmt · 2024-08-12
> > > > >
> > > > > Thank you again for a detailed response.
> > > > >
> > > > > I stand by the points made in my two previous comments, so I keep my updated rating of 3 (Reject). I would like to reiterate my opinion that the technical core of the work has undeniable value, but the submission would need substantial modifications, as was discussed.

---

### Official Review · Reviewer_b2tW · 2024-06-25

**Soundness:** 3
**Presentation:** 4
**Contribution:** 3
**Rating:** 7
**Confidence:** 3

**Summary:**

This work, of a theoretical nature, considers the problem of minimizing a functional $$\mathcal{F}$$ over the space of probability measures of the form

$$\mathcal{F}(\mu) = \int (G(x) - H(x)) d\mu(x) + \mathcal{H}(\mu)$$

where $G,H$ are **convex** potentials and $\mathcal{H}$ is (typically) the negative entropy (the work actually extends to functionals $\mathcal{H}$ that are convex along generalized geodesics).

This setting substantially differs from standard optimization in the space of probability measure in that $\mathcal{F}$ is not convex along generalized geodesics, the standard assumption to establish convergence of gradient flows in this spaces, following the seminal work of Ambrosio, Gigli and Savaré. Nonetheless, the "difference of convex functions" (DC) structure enables the derivation of a two-step scheme to minimize $\mathcal{F}$. Namely, given a current iterate $\mu_n$, one first computes $\nu_{n+1}$ using a **forward** (explicit) Euler scheme _using only the concave term_ $-H$, then obtain $\mu_{n+1}$ using a **backward** (implicit) iterate on the convex (along generalized geodesics) term $\mu \mapsto \braket{G,\mu} + \mathcal{H}(\mu)$.

This approach is referred to as a _semi_ forward-backward (FB) Euler scheme, seen as a variation of the known FB scheme (which would process similarly, but performing an explicit iterate on $G-H$ and an implicit iterate on $\mathcal{H}$).

In contrast to the standard FB scheme, this semi-FB scheme that leverages the DC structure enables the derivation of strong theoretical guarantees: existence of accumulation points that are critical points of $\mathcal{F}$, and under additional standard assumptions ($\sim$ PL inequality), convergence rate toward the global minimum of $\mathcal{F}$.

**Strengths:**

The presentation of the work is extremely clear. It is exposed in a pedagogical way, motivations are clear and results are stated formally.

The authors obtain strong theoretical guarantees in a scope more general than the restricted _convex along generalized geodesics_ one. The DC framework in the context of Wasserstein-based optimization seems to be a promising playground for further research.

The proposed approach is elegant to me, in that it is simple yet seems very powerful, and seems reasonably usable in practice.

**Weaknesses:**

1. While I understand that the paper is mostly theoretical, the "from theory to numeric" side may be enhanced (typically, putting some of the material of Part B, in particular Algorithm 2, to the main body).

2. Still targeting numerical applications, the approach seems somewhat limited in that the numerical experiments, according to the appendix, have been run on a fairly powerful hardware and yet took a couple of hours to run, while they remain at the proof-of-concept level on toy datasets.

3. Though this does not diminishes the intrinsic quality of the paper, the paper relies on technical proofs deferred to the appendix and I could not proofread all of them. I read the proof of Lemma 1 and Theorem 1, which seem correct to me, and quickly check other proofs but cannot guarantee the correctness of the theorems (though they look sounded)---which are nonetheless the main (if not unique) contributions of the work. I understand that this is not the authors fault, but I wanted to stress that point in my review.

**Minor aspects:**
- It may be better to use $\tau$ instead of $\gamma$ to denote the step-size in JKO schemes, since $\gamma$ is often used to denote a transport plan.
- I believe it's better to avoid starting sentences with mathematical symbol. That makes the document a bit easier to read. See for instance line 117 "continuous w.r.t. $\mathcal{L}^d$. $\mu$-a.e. stands..." is a bit confusing.

**Questions:**

1. Genuine question: is there any convergence in the regime $\gamma \to 0$ toward a limit curve $(\mu_t)_t$, and if so does it coincides with the usual Wasserstein gradient flow of $\mathcal{F}$? Said differently, is it true that the (time-continuous) Wasserstein gradient flow of the KL for a log-DC target distribution converges globally? (As a consequence of the fact that the sequence $(\mu_n)_n$ you build do converge globally in that case)

---

> ### Author Rebuttal · Authors · 2024-08-06
>
> We thank the reviewer for your positive comments on our work.
>
> **Weaknesses**
>
> ***[1.]*** We will put some material from Part B (discussion on the ICNN approach for the JKO as well as Algorithm 2) into the main text as suggested when we have some extra space.
>
> ***[2.]*** We agree with this limitation. Please see also our reply to reviewer 7EW7. As answered to reviewer 7EW7, there are some recent works [A-Melis22, Fan22] proposing new strategies to compute the JKO that scale better.
>
> *[A-Melis22] Alvarez-Melis, D., Schiff, Y., & Mroueh, Y. (2022). Optimizing functionals on the space of probabilities with input convex neural networks. Transactions on Machine Learning Research.*
>
> *[Fan22] Fan, J., Zhang, Q., Taghvaei, A., & Chen, Y. (2022). Variational Wasserstein gradient flow. International Conference on Machine Learning.*
>
> ***[3.]*** As authors, we also try our best to ensure everything is mathematically correct.
>
>
> ***Minor aspect.*** Thanks for the suggestion. We will replace $\gamma$ with some other symbol and also revise all the sentences carefully.
>
> ***Question.*** It is sensible to expect such behaviours of the time-continuous limit of semi-FB-Euler. We think the hypotheses raised by the reviewers are very likely to hold because the semi-FB-Euler is stable and has sufficient regularity. However, we have not really studied that limit and it is not necessarily easy to prove such hypotheses. This could be an interesting future research avenue.

---

> > ### Comment · Reviewer_b2tW · 2024-08-08
> > **Thanks**
> >
> > Thank you for taking time answering my review :-)

---

### Official Review · Reviewer_TQTP · 2024-07-11

**Soundness:** 3
**Presentation:** 3
**Contribution:** 3
**Rating:** 7
**Confidence:** 4

**Summary:**

This work focuses on the optimization in the Wasserstein space of a functional which is a sum of an internal energy (convex along a generalized geodesic) and of a potential energy, whose potential is a difference of convex functions. To solve such problem, the authors propose to generalize the semi Forward-Backard Euler scheme to the Wasserstein space, and study it theoretically, showing the convergence in general settings. Finally, they demonstrate the convergence of the algorithm on two toy objectives.

**Strengths:**

- This paper is well written and clear.
- To minimize functionals obtained as difference of convex functionals, it introduces the semi-forward backward Euler scheme, and it shows its convergence under different reasonable assumptions.
- The theoretical analysis seems rigorous and complete, as it shows the convergence in different settings and under different criteria (e.g. the paper provides a descent lemma, convergence of a subsequence towards a critical point, convergence in gradient, and convergence to the minimum under PL inequality).

**Weaknesses:**

This is mainly a theoretical work. Thus, the following weaknesses on the experiment section and on the method are minor in my opinion.
- As underlined in the experiment section, the scheme involves to compute the JKO operator, which is computationally costly as it requires neural networks to solve it
- The experiments focus on 2D toy examples

**Questions:**

The focus of the paper is on functionals obtained as a sum of a potential and an internal energy. Could the analysis be done with more general DC functionals (e.g. with interaction energies.). I guess yes since the MMD is presented in the introduction as being such a functional. It would also have been nice to test the semi Forward-Backward scheme on the MMD, as it is not geodesically convex, and does not converge well without tricks such as adding noise (e.g. for Gaussian kernel).
Assumption 2 supposes that the sublevel sets are compact w.r.t the Wasserstein topology. Are there easy examples of functionals which satisfy this (e.g. those used in the experiment section)?

Typos:
- In the abstract, I found the sentence line 3 "When the regularization term is the negative entropy, the optimization problem becomes a sampling problem where it minimizes the Kullback-Leibler divergence between a probability measure (optimization variable) and a target probability measure whose logarithmic probability density is a nonconvex function." weird as it is not specified that the objective function is a potential.
- Line 14: "can be considered gradient descent": lack a word?
Rating: 7

**Limitations:**

The authors have acknowledged the limitations in the main text

---

> ### Author Rebuttal · Authors · 2024-08-06
>
> We thank the reviewer for constructive and positive comments on our work.
>
> **Weaknesses**
>
> ***the scheme involves to compute the JKO operator, which is computationally costly..., toy examples***: Please also refer to our reply to reviewer 7EW7. Since we use [Mokrov21] in our JKO step, our work is equally costly. There are some recent works [A-Melis22, Fan22] that propose new strategies to compute the JKO that scale better.
>
> *[Mokrov21] Mokrov, P., Korotin, A., Li, L., Genevay, A., Solomon, J. M., & Burnaev, E. (2021). Large-scale Wasserstein gradient flows. Advances in Neural Information Processing Systems, 34, 15243-15256.*
>
> *[A-Melis22] Alvarez-Melis, D., Schiff, Y., & Mroueh, Y. (2022). Optimizing functionals on the space of probabilities with input convex neural networks. Transactions on Machine Learning Research.*
>
> *[Fan22] Fan, J., Zhang, Q., Taghvaei, A., & Chen, Y. (2022). Variational Wasserstein gradient flow. International Conference on Machine Learning.*
>
> **Question**
>
> ***``Could the analysis be done with more general DC functionals"***: Yes, our framework can handle interaction energies (e.g., MMD) as discussed in the context paragraph and Appendix A.2. Applying semi-FB Euler to MMD as suggested is a good research direction which we can consider in the near future.
>
> ***``Assumption 2 supposes that the sublevel sets are compact..."***: As far as we know, there is no easy check for it when the base space for the Wasserstein space is $\mathbb{R}^n$. If the base space is instead compact ($X \subset  \mathbb{R}^n$, $X$ is compact, the Wasserstein space is the space of all finite 2nd-moment probability distributions whose supports are contained in $X$), the compactness of sublevel sets would be easier to verify since the coercivity of $\mathcal{F}$ is a sufficient condition. In turn, to check coercivity, one sufficient condition is that the potential grows fast enough at infinity, e.g., if $\mathscr{H}$ is the entropy, we need $F(x) \gtrsim c \Vert x \Vert^2$. Nevertheless, this is not the case here since we wanted to consider $\mathbb{R}^n$ but not $X$.
>
> Note, however, that the compactness assumption is only used for Theorem 1. In fact, we only need the compactness of the generated sequence {$\mu_n$} in the proof. Therefore, in the final version, we will remove Assumption 2 and instead assume that the sequence {$\mu_n$}  is compact inside Theorem 1.
>
> ***Typos***: We will proofread the paper and correct the typos as suggested.

---

### Official Review · Reviewer_7EW7 · 2024-07-15

**Soundness:** 3
**Presentation:** 3
**Contribution:** 3
**Rating:** 6
**Confidence:** 3

**Summary:**

This paper studies minimization algorithms for functionals on the Wasserstein space with the following difference of convex functions (DC) structure:
$$
\min_{\mu \in \mathcal{P}_2(X)} \mathcal{F}(\mu) := \int (G(x) - H(x)) d\mu(x) + \mathcal{H}(\mu),
$$ where $G$ and $H$ are convex functions and $\mathcal{H}$ is convex along (generalized) geodesics in the Wasserstein space. The main motivation for this problem is that it is the sampling analog to difference of convex functions problems in optimization, giving a problem of intermediate difficulty between convex and true non-convex. The authors also mention an application to maximum mean discrepancy.

They study a modified forward Euler sceme which they refer to as "semi FB Euler", where given a current iterate $\mu_n$ they update as
$$
\mu_{n + 1/2} := (I + \gamma \nabla H) \mu_n,
$$ and then
$$
\mu_{n + 1} := JKO_{\gamma(H+ E_G)}(\mu_{n + 1/2}).
$$ A similar scheme was considered in [1,2] but with the difference that the function $G$ was included in the first step but not in the second; the motivation for this difference is that it more closely aligns with DC programming. One benefit of this modification is that $\mu_{n + 1/2}$ is pushed forward by an optimal transport map, which makes the problem more tractable.

The main focus of the work are various convergence results for the above scheme, including an asymptotic result as well as results on the norm of a gradient analog and the distance between $0$ and the sub-differential. The authors also consider convergence under a gradient domination condition, and conclude with some numerics.

[1] Wibisono, Andre. "Sampling as optimization in the space of measures: The Langevin dynamics as a composite optimization problem." Conference on Learning Theory. PMLR, 2018.

[2] Salim, Adil, Anna Korba, and Giulia Luise. "The Wasserstein proximal gradient algorithm." Advances in Neural Information Processing Systems 33 (2020): 12356-12366.

**Strengths:**

- This paper is, to my knowledge, the first to consider the difference of convex functions setting.

- The above bullet leads the authors to propose a modified algorithm compared to previous literature, and means that their convergence results are generally novel.

- The work is rigorous and quite mathematically clear.

**Weaknesses:**

- The Łojasiewicz condition is not verified to hold in any settings other than their the standard case of KL minimization.

- The problem is somewhat far from practice, firstly because the JKO step is very challenging to implement in practice, and secondly because the setting the authors study is only motivated by one application, to maximum mean discrepancy functionals. They do mention a connection to two-layer neural networks in the "Context" section on page 2 but this was impossible to understand from the little they wrote.

- The proofs don't yield explict constants in their results, which is important because there could be hidden dimension-dependence.

**Questions:**

- The thorough introduction to Wasserstein space is much appreciated, but perhaps it could be largely moved to an appendix so that more space is left in the main text for more discussion of applications and high-level ideas of the proofs.

- Please give more space to the connection to two-layer neural networks mentioned in the "Context" section on page 2.

- Since it is a general fact (see, e.g. [1]) that a gradient domination condition implies that $d(x, x*)^2 <= C(F(x) - F(x*))$, I feel that Theorem 5 should either be removed, or stated as a corollary in an appendix and just mentioned as a brief remark in the main text.

[1] Otto, Felix, and Cédric Villani. "Generalization of an inequality by Talagrand and links with the logarithmic Sobolev inequality." Journal of Functional Analysis 173.2 (2000): 361-400.

**Limitations:**

The authors have adequately addressed the limitations.

---

> ### Author Rebuttal · Authors · 2024-08-06
>
> We thank the reviewer for the thoughtful and positive comments on our work.
>
> **Weaknesses**
>
> ***$\bullet$ "The Lojasiewicz condition..."***: We agree that the Lojasiewicz condition is usually used in the case of KL divergence. However, the condition also almost holds for the case of Maximum Mean Discrepancy (MMD) (under mild assumptions). Recall that MMD involves the interaction energy:
>
> \begin{align*}
>     \mathcal{F}(\mu) = \dfrac{1}{2} D_{\text{MMD}}(\mu^*,\mu)^2:=\dfrac{1}{2} \int \int k(x,y) d\mu(x) d\mu(y) + \int{F(x)} d\mu(x) + C
> \end{align*}
> where $D_{\text{MMD}}$ is the MMD distance, $F(x)=-\int{k(x,y)}d\mu^*(y)$, $C$ does not depend on $\mu$ and $k$ is a stationary kernel, i.e., $k(x,y) = W(x-y)$ for some $W$. The Wasserstein gradient of $\mathcal{F}$ is given by
> \begin{align*}
>     \nabla_W \mathcal{F}(\mu) = \nabla F + \nabla W * \mu
> \end{align*}
> where * denotes the convolution. Under some regularity conditions of the kernel, it is known that (inequality (62) in [Arbel19] with some notational change)
> \begin{align*}
>     2(\mathcal{F}(\mu)-\mathcal{F}(\mu^*)) \leq \Vert \mu^*-\mu\Vert_{\dot{H}^{-1}(\mu)} \times \int{\Vert \nabla_W \mathcal{F}(\mu) \Vert^2} d\mu(x)
> \end{align*}
> where $\Vert \mu^*-\mu\Vert_{\dot{H}^{-1}(\mu)}$ is the weighted negative Sobolev distance. This is almost the Lojasiewicz condition for $\mathcal{F}$. The only requirement to make it real Lojasiewicz is that $\Vert \mu-\mu^*\Vert_{\dot{H}^{-1}(\mu)}$ is bounded uniformly for all $\mu$. However, this is tricky. Nevertheless, the point is that we only use the Lojasiewicz condition for the generated sequence {$\mu_k$}, so all we need is the above quantity to be bounded along that sequence, and this boundedness seems natural and is usually **assumed** (as in [Prop.7, Arbel19]). We will add this discussion in the revised version.
>
> *[Arbel19] Arbel, M., Korba, A., Salim, A., & Gretton, A. (2019). Maximum mean discrepancy gradient flow. Advances in Neural Information Processing Systems, 32.*
>
> ***$\bullet$ "JKO step is very challenging"***: The bottleneck of the JKO computation using [Mokrov21] is the logdet(Hessian), (scales cubically w.r.t. data dimension) and training an NN at each step. If we trade some accuracy for scalability, we can use stochastic log determinant estimators as in [A-Melis22], leading to quadratic complexity. We are also aware of a recent work that replaces the gradient of ICNN by a residual neural network, leading to remarkable speeding [Fan22]. Therefore, we believe the scalability issue of the JKO and hence our proposed method can be resolved shortly.
>
> *[Mokrov21] Mokrov, P., Korotin, A., Li, L., Genevay, A., Solomon, J. M., & Burnaev, E. (2021). Large-scale Wasserstein gradient flows. Advances in Neural Information Processing Systems, 34, 15243-15256.*
>
> *[A-Melis22] Alvarez-Melis, D., Schiff, Y., & Mroueh, Y. (2022). Optimizing functionals on the space of probabilities with input convex neural networks. Transactions on Machine Learning Research.*
>
> *[Fan22] Fan, J., Zhang, Q., Taghvaei, A., & Chen, Y. (2022). Variational Wasserstein gradient flow. ICML.*
>
> ***--``only motivated by one application, to maximum mean discrepancy functionals"***: our work is not only motivated by maximum mean discrepancy but also (and mainly) by sampling from log-DC densities, which existing samplers fail to address. This is a very rich class of (nonconvex) densities and it would be helpful to have another independent survey on DC structures of commonly-used densities.
>
> ***--"connection to two-layer neural networks"***: The connection between MMD and the problem of infinite-width one hidden layer neural network is briefly mentioned in [Salim20, Appendix B.2]. We can add a pointer to that and can even briefly present it for completeness.
>
> *[Salim20] Salim, A., Korba, A., & Luise, G. (2020). The Wasserstein proximal gradient algorithm. Advances in Neural Information Processing Systems, 33, 12356-12366.*
>
> ***$\bullet$``The proofs don't yield explicit constants"***: we can discuss the complexity w.r.t. dimensionality (to some extent). However, yielding explicit dimension-constant is very tricky, especially since the ICNN family is only a subfamily of convex functions, and we do not know how close the solution given by ICNN is to the true optimal transport map. So, obtaining the overall complexity that includes the JKO computation is probably impossible in practice.
>
> **Question**
>
> ***$\bullet$ ``The thorough introduction..."***: Thanks for the suggestion. However, we think it would be beneficial for the readers to have Wasserstein's geometry background in the main text, and we also need all those notions to present our results precisely, so moving them to the appendix is a bit difficult. In the final version with one page extra, we will extend the experiment section, e.g., move the practical scheme in the appendix to the main text.
>
> ***$\bullet$``Please give more space..."***: as answered above, we can add a pointer to [Salim20, Appendix B.2] or can briefly discuss it for completeness.
>
> ***$\bullet$ ``Since it is a general fact..."***: It is known that Log-Sobolev implies Talagrand [Otto00]; consequently, KL divergence controls Wasserstein distance in such a case. We can discuss this connection in our paper. However, Theorem 5 is much more general than that and cannot use the above implication. First, we work with a general regularizer $\mathscr{H}$, so the objective is not KL divergence. Second, even when $\mathscr{H}$ is the negative entropy, resulting in KL divergence, we consider the general Lojasiewicz exponent $\theta \in [0,1)$. At the same time, the above implication applies only for $\theta=1/2$ (the case of Log-Sobolev inequality). Therefore, Theorem 5 is not simply a corollary of Theorem 4.
>
> *[Otto00] Otto Felix and Cédric Villani. "Generalization of an inequality by Talagrand and links with the logarithmic Sobolev inequality." Journal of Functional Analysis 173.2 (2000): 361-400*

---

> > ### Comment · Reviewer_7EW7 · 2024-08-12
> > **Thanks for your response**
> >
> > With regard to the Lojasiewicz condition, it is interesting that it can almost be satisfied in the MMD case, but I would emphasize that the reason the log-Sobolev inequality is so important is that it holds uniformly. In particular, the gap between a PL inequality at some points versus a PL inequality everywhere can be very large and difficult to bridge. Also, I still don't find the connection to MMD particularly convincing, so this setting is not completely convincing anyways.
> >
> > With regard to the fact that gradient domination implies quadratic growth: sure, the case where $(f(x) - f(x_*))^\theta \leqslant C\|\nabla f(x)\|$  is not explicitly proved in [1], but the proof can be quite easily modified to handle that case. At a minimum, you should add citations to [1] and [2]. Note also that using their results you could likely simplify your proofs and make the constants more explicit.
> >
> > With regard to your remaining points, thanks for your responses. Overall, my opinion of the work has not substantially changed, so I leave my score as-is.
> >
> > [1] Otto, Felix, and Cédric Villani. "Generalization of an inequality by Talagrand and links with the logarithmic Sobolev inequality." Journal of Functional Analysis 173.2 (2000): 361-400.
> > [2] Karimi, Hamed, Julie Nutini, and Mark Schmidt. "Linear convergence of gradient and proximal-gradient methods under the polyak-łojasiewicz condition." Machine Learning and Knowledge Discovery in Databases: European Conference, ECML PKDD 2016, Riva del Garda, Italy, September 19-23, 2016, Proceedings, Part I 16. Springer International Publishing, 2016.

---

> > > ### Author Response · Authors · 2024-08-13
> > >
> > > Thank you for the response.
> > >
> > > We will incorporate those suggestions into the final version of the paper.

---

### Author Rebuttal · Authors · 2024-08-06

We thank all the reviewers for their thoughtful, constructive, and high-quality reviews. We reply to each reviewer in their comment section.

---

### Decision · Program_Chairs · 2024-09-25

**Decision:**

Accept (poster)

**Comment:**

The paper takes a stab at DC optimization in the Wasserstein space and proposes a semi Forward-Backard Euler scheme (section 3, equation 4). All the reviewers like the work especially as it offers a fresh perspective on the problem. Hence, we have decided to accept the work. Please incorporate all the asked changes.

One reviewer had comments on whether the developments are complete and suggested some explorations. Although it would not be possible to accommodate such detailed explorations in the final version, I would still encourage the authors to provide a short commentary on the future directions / limitations.